# A Provable Expressiveness Hierarchy in Hybrid Linear-Full Attention

Xiaowei Ye [* 1]   Xiaoyu He [* 2]   Chao Liao [2]   Chen Wu [2]   Pinyan Lu [2 3]

## Abstract

Transformers serve as the foundation of most modern large language models. To mitigate the quadratic complexity of standard full attention, various efficient attention mechanisms, such as linear and hybrid attention, have been developed. A fundamental gap remains: their expressive power relative to full attention lacks a rigorous theoretical characterization. In this work, we theoretically characterize the performance differences among these attention mechanisms. Our theory applies to all linear attention variants that can be formulated as a recurrence, including Mamba, DeltaNet, etc. Specifically, we establish an expressiveness hierarchy: for the sequential function composition-a multi-step reasoning task that must occur within a model's forward pass, an $(L + 1)$-layer full attention network is sufficient, whereas any hybrid network interleaving $L - 1$ layers of full attention with a substantially larger number $(2^{3L^2})$ of linear attention layers cannot solve it. This result demonstrates a clear separation in expressive power between the two types of attention. Our work provides the first provable separation between hybrid attention and standard full attention, offering a theoretical perspective for understanding the fundamental capabilities and limitations of different attention mechanisms.

## 1. Introduction

The Transformer architecture (Vaswani et al., 2017) serves as the backbone of modern large language models (LLMs), exemplified by numerous recent models (OpenAI, 2023; DeepSeek-AI, 2024; MiniMax, 2025b; Team et al., 2025;

*Equal contribution [1]École Polytechnique, Palaiseau, France [2]Taylor Lab, Huawei [3]Key Laboratory of Interdisciplinary Research of Computation and Economics, Shanghai University of Finance and Economics, Shanghai, China. Correspondence to: Xiaoyu He is now at MiniMax <heyu@minimaxi.com>, Pinyan Lu <lu.pinyan@mail.shufe.edu.cn>.

*Proceedings of the 43^{rd} International Conference on Machine Learning*, Seoul, South Korea. PMLR 306, 2026. Copyright 2026 by the author(s).

Yang et al., 2025a). Its core component is the *self-attention unit*: in the standard full attention mechanism, it models interactions among input elements as inner-products between low-dimensional embeddings and calculates the output via a softmax weighted sum.

Despite the success of standard full attention, its quadratic computation and linear-memory complexity remain significant bottlenecks. Hence, various new attention mechanisms have been proposed, including linear attention (Katharopoulos et al., 2020; Kasai et al., 2021; Schlag et al., 2021; Peng et al., 2021; Yang et al., 2024) used in models such as Mamba (Gu & Dao, 2024), Minimax-M1 (MiniMax, 2025a), RWKV (Peng et al., 2025) and Gated DeltaNet (Yang et al., 2025b); linear-full hybrid attention used in models such as Hunyuan-TurboS (Team, 2025b), Qwen3-Next (QwenTeam, 2025), and Kimi Linear (Team, 2025a)); as well as log-linear attention (Guo et al., 2025) and sparse attention mechanisms (Guo et al., 2019; Qiu et al., 2020; Zaheer et al., 2020; Beltagy et al., 2020; Guo et al., 2022; Yuan et al., 2025; Lu et al., 2025). A natural question arises: *Can these attention mechanisms perform better than full attention?* This paper aims to answer this question through a theoretical comparison of these attention mechanisms with full attention.

We first analyze the Transformer with hybrid attention mechanisms on the sequential function composition task introduced by (Chen et al., 2025). Our results show that even when the number of linear attention layers grows exponentially relative to the number of full attention layers, the performance gain remains marginal.

Moreover, we examine sparse attention mechanisms. We establish a hardness result for the 2-Sum task under a single-layer sparse attention setup, providing the first provable separation between sparse attention and full attention.

### 1.1. Our result

Consider a Transformer with $H$ attention heads, head dimension $d$, precision $p$, and prompt length $n$. We establish the following lower bound for a class of hybrid attention mechanisms.

**Theorem 1.1** (Informal)**.** *Any small hybrid architecture (Definition 2.3) that mixes $L - 1$ full attention layers with*

*exponentially many linear attention layers* **cannot** *solve the L-sequential function composition task (Definition 2.5).*

Our analysis covers a broad class of linear attention mechanisms that admit a recurrent formulation—including Mamba (Gu & Dao, 2024), Minimax-M1 (MiniMax, 2025a), RWKV (Peng et al., 2025) and Gated DeltaNet (Yang et al., 2025b)—establishing a general framework for comparison.

The formal statement of Theorem 1.1 will be given in Theorem 2.6. Together with the findings in (Chen et al., 2025), our result implies that incorporating linear attention layers does not yield substantial performance gains. A detailed comparison is presented in Table 1.

*Table 1.* Complexity of $L$-FuncComp for different architecture.

| Model/Task | $L$-SeqCom |
|---|---|
| Full - $L$ layers | $\Omega(\text{poly } n)$ (Chen et al., 2025) |
| Full - $L+1$ layers | $O(\text{poly} \log n)$ (Chen et al., 2025) |
| Hybrid - $L-1$ full layers | $\Omega(\text{poly } n)$ (Theorem 2.6) |

We remark on the $L$-sequential function composition task, which we discuss in this paper. This task formally captures the essence of multi-step reasoning (e.g., multi-hop retrieval) that must occur within a model's forward pass, where the solution requires composing functions sequentially, with the output of one step defining the input context for the next. Our theoretical instantiation employs carefully constrained "retrieval scopes" at each step to enable rigorous analysis. We discuss the choice of the tasks in Appendix E. This task is of a theoretical nature, and the implications of our results in the real world demand verification of practice.

We also analyze sparse attention mechanisms.

**Theorem 1.2** (Informal)**.** *There exists a task (2-Sum, Definition 2.7), such that a small single-layer full attention solves it easily, while a small single-layer block-sparse attention mechanism (called $(B, k)$-sparse attention, defined in Definition 2.4)* **cannot** *solve it.*

The formal result is given in Theorem 2.8. This result demonstrates a strict efficiency gap for large $B$. Our proofs adopt a methodology based on communication complexity. We provide an overview of our results in Table 2.

*Table 2.* Complexity of full and sparse attention for 2-Sum.

| Model | Complexity |
|---|---|
| Full | $O(\log n)$ (Sanford et al., 2023) |
| Sparse (Block) | $\Omega(B \log n)$ (Theorem 2.8) |

### 1.2. Organization and overview

Section 2 provides the mathematical background, including formal definitions of attention mechanisms and the studied

tasks: sequential function composition and 2-Sum. Section 3 introduces communication protocol for hybrid attention mecanisms to solve sequential function composition. In Section 4, we adapt methods from (Chen et al., 2025) to derive a lower bound for a hybrid architecture, thus proving Theorem 1.1. Finally, Section 5 establishes Theorem 1.2, offering the theoretical limitation on sparse attention. Detailed proofs will be given in Appendices A, B and D, and we also discuss the limitations of this work in Appendix E.

### 1.3. Related work

**Transformer-RNN comparison.** There have been several results on the theoretical comparison between Transformers and RNNs (Elman, 1990). (Jelassi et al., 2024) demonstrated a representation gap between RNNs and Transformers in repeating a long sequence. (Sanford et al., 2023) proved a linear-constant separation of RNNs and Transformers on the sparse averaging task, and later a separation on $k$-hop induction head tasks (Sanford et al., 2024a).

(Bhattamishra et al., 2024) establishes computational separations between Transformers and RNNs on tasks such as Index Lookup, String Equality, Nearest Neighbor, and Associative Recall. Additionally, they prove a linear lower bound for one-layer Transformers on bounded Dyck languages (Hahn, 2020), forming a separation as constant-size RNNs solve this task (Bhattamishra et al., 2020; Hewitt et al., 2020). In practice, however, one-layer Transformers are rarely used. Multi-layer Transformers are shown to succeed on bounded Dyck languages (Wen et al., 2023; Yao et al., 2021), indicating that Transformers match RNNs on language recognition while excelling at retrieval.

Further work by (Wen et al., 2025) extends the comparison to one-layer RNNs augmented with chain-of-thought (CoT). Although CoT improves RNN performance, it fails to close the representational gap with Transformers: under mild complexity assumptions, an RNN with CoT is strictly more powerful than a plain RNN but still exponentially weaker than a CoT-enhanced Transformer on algorithmic problems. (Wen et al., 2025) also suggest that RAG-based improvements can achieve Turing completeness, offering ways to narrow the gap. One proposed approach uses a hybrid RNN-Transformer mechanism, which has been implemented in models like Hunyuan-TurboS (Team, 2025b), Qwen3-Next (QwenTeam, 2025), and Kimi Linear (Team, 2025a).

We make some contributions to the comparative expressiveness of recurrent and transformer-based models. In particular, we establish a new separation result for hybrid architectures on the sequential function composition task.

**Theoretical study of Transformer.** Theoretical research on Transformers has progressed along two main fronts: expressive power and inherent limitations. The line on expressive

power began with (Pérez et al., 2019)'s investigation of their ability to emulate Turing machines, followed by studies on adversarial robustness (Hsieh et al., 2019), universal approximation (Yun et al., 2020; Wei et al., 2022a), and Turing completeness (Pérez et al., 2021; Wei et al., 2022a).

Concerning limitations, (Hahn, 2020) pioneered this direction by proving that hard-attention Transformers cannot recognize parity or Dyck languages. Subsequent work established further limitations—either via communication complexity for one-layer model (Peng et al., 2024; Sanford et al., 2023; 2024a;b) or under circuit complexity conjectures (Sanford et al., 2024a; Peng et al., 2024; Merrill & Sabharwal, 2023). A significant advance was made by (Chen et al., 2025), who proved the first unconditional lower bound for multi-layer Transformers (on sequential function composition) by introducing the concept of indistinguishable decomposition. In this paper, we adapt their techniques to derive a lower bound for hybrid architectures.

**Chain of Thought (CoT).** Chain-of-Thought (CoT) (Wei et al., 2022b) enhances the reasoning by inducing step-by-step reasoning traces, thereby providing Transformers with an augmented computational workspace. This augmentation confers significant expressive benefits: constant-size Transformers with CoT are known to simulate any polynomial-time algorithm (Pérez et al., 2021; Merrill & Sabharwal, 2024; Feng et al., 2023; Li et al., 2024; Li & Wang, 2025). Given that Transformers (without CoT) are believed to be limited to the complexity class $\mathsf{TC}^0$ (Merrill & Sabharwal, 2023), these results suggest (assuming standard conjectures like $\mathsf{P} \not\subset \mathsf{TC}^0$) that CoT strictly improves the computational power of Transformers. Notably, (Chen et al., 2025) recently proved the first unconditional separation for CoT, introducing a key proof technique for such results. They prove that a single-layer small Transformer with CoT can solve the $L$-sequential function composition task, while an $L$-layer small Transformer cannot.

Complementary to these expressiveness results, another line of work establishes lower bounds on the number of CoT steps necessary for specific tasks. Lower bounds have been shown for single-layer Transformers on iterated function composition (Peng et al., 2024). Other works connect the required step count to structural complexity measures like the Ehrenfeucht-Haussler rank (Barcelo et al., 2025) or establish bounds for various problems (e.g., parity, multiplication, median, graph reachability) under restricted attention patterns (Amiri et al., 2025).

**Sparse attention mechanisms.** The quadratic complexity of full attention creates a significant computation bottleneck for long-context processing. Sparse attention mechanisms, leveraging inherent sparsity in attention matrices (Ge et al., 2024; Jiang et al., 2023), have emerged as a key approach to improve efficiency. Such mechanisms have been applied in

many models such as MoBA (Lu et al., 2025), NSA (Yuan et al., 2025), DSA (DeepSeek-AI, 2025), etc.

Despite their widespread adoption for computational efficiency, we establish a fundamental limitation of (single-layer) sparse attention: on tasks requiring uniform attention across all input tokens, any sparse mechanism is provably less powerful than full attention. We provide a formal lower bound for compression- and selection-based sparse strategies, which constitute two of the three canonical categories outlined by (Yuan et al., 2025) (the third, sliding window, is functionally analogous to an RNN layer).

## 2. Preliminaries

We begin by formalizing the key components of our study. This section first introduces the attention mechanisms we analyze: full attention, linear attention, log-linear attention, and hybrid architectures. We then formally define the primary tasks: function evaluation, permutation composition, and sequential function composition.

**Notation.** For integers $n \geq 0$ and $n_2 \geq n_1$, we use the notation $[n] = \{1, 2, \ldots, n\}$ and $[n_1 : n_2] = \{n_1, n_1 + 1, \ldots, n_2\}$, with the convention $[0] = \varnothing$.

### 2.1. Transformer with full attention

We consider the decoder-only Transformer architecture. Following the notations of (Chen et al., 2025), let $L$ be the number of attention layers, $H$ be the number of attention heads per layer, $p$ be the precision, $d$ be the model dimension, and $n$ be the (input) prompt length. It is typically assumed that $Hdp \geq O(\log(n))$. An $L$-layer decoder-only Transformer consists of alternating attention layers and MLP layers:

$$f_{\mathsf{tran}} = f_{\mathsf{mlp}}^{(L)} \circ f_{\mathsf{attn}}^{(L)} \circ \cdots \circ f_{\mathsf{mlp}}^{(1)} \circ f_{\mathsf{attn}}^{(1)}$$

Given an input sequence $x^{(0)} = (x_1^{(0)}, \ldots, x_n^{(0)}) \in (\mathbb{R}^{dH})^n$, the Transformer inductively computes the output of the $\ell$-th attention layer $y^{(\ell)} = (y_1^{(\ell)}, \ldots, y_n^{(\ell)})$ and the output of the $\ell$-th MLP layer $x^{(\ell)} = (x_1^{(\ell)}, \ldots, x_n^{(\ell)})$. For layer $\ell \in [L]$.

• **Attention layer** $f_{\mathsf{attn}}^{(\ell)}$: For each attention head $h \in [H]$ and position $i \in [n]$, the output is computed as

$$y_i^{(\ell, h)} = \sum_{j \leq i} \alpha_{i,j}^{(\ell, h)} V^{(\ell, h)} x_j^{(\ell-1)} \in \mathbb{R}^d \qquad (1)$$

where $\{\alpha_{i,j}^{(\ell, h)}\}_{j \leq i}$ is the attention score from the $h$-th head, given by the softmax operation

$$\alpha_{i,j}^{(\ell, h)} = \frac{\exp((x_i^{(\ell-1)})^\top (Q^{(\ell, h)})^\top K^{(\ell, h)} x_j^{(\ell-1)})}{\sum\limits_{j \leq i} \exp((x_i^{(\ell-1)})^\top (Q^{(\ell, h)})^\top K^{(\ell, h)} x_j^{(\ell-1)})}.$$

Here, $Q^{(\ell,h)}, K^{(\ell,h)}, V^{(\ell,h)} \in \mathbb{R}^{d \times dH}$ denote the query, key, and value matrices of the $h$-th head, with each entry represented using $p$-bit precision.

Finally, the output of the $\ell$-th attention layer is the concatenation of each head,

$$y_i^{(\ell)} = (y_i^{(\ell,1)}, \ldots, y_i^{(\ell,H)}) \in \mathbb{R}^{dH} \quad \forall i \in [n]$$

• **MLP layer** $f_{\mathsf{mlp}}^{(\ell)}$: The output of the $\ell$-th layer (and also the input to the $(\ell+1)$-th layer) is an arbitrary function $g^{(\ell)} : \mathbb{R}^{2dH} \to \mathbb{R}^{dH}$ applied position-wise:

$$x_i^{(\ell)} = g^{(\ell)}(x_i^{(\ell-1)}, y_i^{(\ell)}) \in \mathbb{R}^{dH}.$$

Note that here we modify the definition of the MLP layer to adapt to various types of transformer architectures and residual connections (Zhu et al., 2025; Xie et al., 2025).

## 2.2. Linear attention, RNN, and hybrid attention

For linear attention, the attention probabilities become

$$\alpha_{i,j}^{(\ell,h)} = \frac{\varphi(Q^{(\ell,h)}x_i^{(\ell-1)})^\top \varphi(K^{(\ell,h)}x_j^{(\ell-1)})}{\sum_{j \leq i} \varphi(Q^{(\ell,h)}x_i^{(\ell-1)})^\top \varphi(K^{(\ell,h)}x_j^{(\ell-1)})}$$

where $\varphi : \mathbb{R}^d \to \mathbb{R}^d$ is an arbitrary function. This formulation of linear attention leads to a linear runtime complexity. The key is to suppose that we maintain cumulative states

$$S_i^{(\ell,h)} = S_{i-1}^{(\ell,h)} + V^{(\ell,h)}x_i^{(\ell-1)} \otimes \varphi(K^{(\ell,h)}x_i^{(\ell-1)}),$$
$$Z_i^{(\ell,h)} = Z_{i-1}^{(\ell,h)} + \varphi(K^{(\ell,h)}x_i^{(\ell-1)})$$

with $S_0^{(\ell,h)} = 0$ and $Z_0^{(\ell,h)} = 0$, we have

$$y_i^{(\ell,h)} = \frac{\varphi(Q^{(\ell,h)}x_i^{(\ell-1)})^\top S_i^{(\ell,h)}}{\varphi(Q^{(\ell,h)}x_i^{(\ell-1)})^\top Z_i^{(\ell,h)}}.$$

Indeed, linear attention can be viewed as an RNN.

**Definition 2.1** (Recurrent neural network (RNN)). An RNN layer takes as input the sequence $x = (x_1, \cdots, x_n) \in (\mathbb{R}^d)^n$ and produces an output sequence $y = (y_1, \ldots, y_n) \in (\mathbb{R}^d)^n$ computed as follows. One chooses $\mathrm{h}_0 \in \mathbb{R}^m$, then computes inductively $\mathrm{h}_i = g_{(i)}(x_i, \mathrm{h}_{i-1})$ and $y_i = f_{(i)}(x_i, \mathrm{h}_i)$ for $i = 1, \ldots, n$, and $g_{(i)} : \mathbb{R}^{d+m} \to \mathbb{R}^d$ and $f_{(i)} : \mathbb{R}^{d+m} \to \mathbb{R}^d$ are arbitrary functions.

These $\mathrm{h}_i$ are called hidden states of the RNN layer. Two key factors characterize the capacity of an RNN: the hidden dimension $m$, as defined in the Definition 2.1, and the precision $p$, meaning that $h_i$ are represented by $p$-bit numbers.

**Lemma 2.2** (Linear attention as RNN). *A one-layer and one-head linear attention of dimension $d$ and precision $p$* *can be viewed as an RNN layer of hidden dimension $d^2 + d$ and precision $p$. More generally, linear attention of head number $H$, layer number $L$, dimension $d$, and precision $p$ can be viewed as a multi-head and multi-layer RNN of layer number $L$, hidden dimension $H(d^2 + d)$, and precision $p$.*

The hybrid attention architecture is defined as follows.

**Definition 2.3** (Hybrid Transformer architecture). An $(L, a_1, \cdots, a_L)$-hybrid Transformer is an $(L + a_1 + \cdots + a_L)$-layer Transformer consisting of $L$ full attention layers, each followed by $a_1, \cdots, a_L$ layers of linear attention, respectively. i.e.,

$$(T_{linear}^{(L,a_L)} \circ \cdots \circ T_{linear}^{(L,1)} \circ T_{softmax}^{(L)})$$
$$\circ \cdots \circ (T_{linear}^{(1,a_1)} \circ \cdots \circ T_{linear}^{(1,1)} \circ T_{softmax}^{(1)}).$$

## 2.3. Sparse attention

We consider the sparse attention mechanism with block compression and selection strategies.

**Definition 2.4.** A single-layer one-head $(B, k)$-sparse attention is defined as follows: the input tokens are divided into blocks of $B$ tokens, and then applies a compression map $f : (\mathbb{R}^d)^B \to \mathbb{R}^d$ to get compressed tokens $x_{[1:B]}, \cdots, x_{[tB+1,(t+1)B]} \in \mathbb{R}^d$, where $t = \lfloor \frac{i}{B} \rfloor$.

A block selection function $g : \mathbb{R}^d \times \mathbb{R}^d \to \mathbb{R}$ is then applied to assign a score to each block relative to the current token. The $k$ blocks with the highest scores are selected. Let $s_1, \cdots, s_k$ be the indices of these selected blocks, corresponding to token ranges $[s_1 B : (s_1 + 1)B], \cdots, [s_k B : (s_k + 1)B]$. We then calculate the compressed output and the selected output

$$y_i^{compress} = \sum_{(j+1)B \leq i} \alpha_{i,j} V x_{[jB+1:(j+1)B]}$$
$$y_i^{select} = \sum_{k' \in [k]} \sum_{j \in [s_{k'}B+1:(s_{k'}+1)B]} \alpha_{i,j} V x_{[jB+1:(j+1)B]}$$

and the final output is computed as a weighted combination

$$y_i = \lambda y_i^{compress} + (1 - \lambda) y_i^{select}.$$

## 2.4. Tasks

We formally define the tasks analyzed in this paper: sequential function composition (Chen et al., 2025) and 2-Sum.

**Definition 2.5** ($L$-sequential function composition). Given an integer $L \geq 2$, an $L$-sequential function composition task, denoted $L$-FuncComp$(w, z_0, z_1, \ldots, z_L)$ is defined by positive integers $m, n_1, \cdots, n_{L-1}$, a sequence of functions $z_0, z_1 \ldots, z_L$ and a query $w = (w_1, \ldots, w_{L-1}) \in [n_1] \times \cdots \times [n_{L-1}]$. Here, $z_0 \in [m]$ is the initial input, and $z_\ell \in$

$A_\ell := \{[N_{\ell-1}] \to [N_{\ell-1}]\} \simeq [N_{\ell-1}^{N_{\ell-1}}]$ for $\ell \in [L]$ with

$$N_\ell = m \cdot \prod_{\ell' \in [\ell]} n_\ell \quad \forall \ell \in [0 : L-1]. \tag{2}$$

Compute $i_0 = z_0 \in [m], i_1 = z_1(i_0) \in [N_0]$ and inductively for $\ell = 1, 2, \ldots, L-1$: $i_{\ell+1} = z_{\ell+1}(w_\ell, i_\ell) \in [N_\ell]$. The final output is $L\text{-FuncComp}(w, z_0, z_1, \ldots, z_L) = i_L$.

To solve the $L$-sequential function composition task, we assume the Transformer receives the input prompt in the following format: first, the $L$ functions $z_L, z_{L-1}, \ldots, z_0$ are listed, followed by the query $w$. For simplicity, we assume each entry of a function $z_\ell$ (for $\ell \in [0 : L-1]$) is encoded by a single token (thus requiring $N_{\ell-1}$ tokens for $z_\ell$), and the query $w$ is also encoded in a single token.

Theorem 1.1 then formally states as follows.

**Theorem 2.6.** *For any $L$, an $(L-1, 2^{3L^2}, \cdots, 2^{3L^2})$-hybrid Transformer cannot solve $L$-sequential function composition whenever $Hdp \leq n^{2^{-4L-2}}$.*

**Definition 2.7** (The 2-Sum task). Given input sequence $(x_i)_{i \in [n+1]} \in [M]^{n+1}$, with $M = O(n)$, the goal is to output the sequence $(y_i)_{i \in [n]}$ with

$$y_i = \begin{cases} 1, & \exists j < i, x_i + x_j \equiv 0 \mod M \\ 0, & \text{otherwise} \end{cases}.$$

Theorem 1.2 then formally states as follows.

**Theorem 2.8.** *Any single-layer $(B, k)$-sparse attention cannot solve 2-Sum unless $Hdp = \Omega(B \log n)$, while a full attention with $L = 1, H = 1, d = 3$, and $p = \log n$ can.*

## 3. Hybrid communication model

To prove Theorem 2.6, we introduce an $(L, a_1, \cdots, a_L)$-hybrid communication model for $L$-sequential function composition, which extends the framework of (Chen et al., 2025). The model comprises $L + 2$ players, each holding one of the following: the $L$ functions, the initial input, or the query. Communication is organized into $L$ epochs. Within each epoch, a single round simulates a full attention layer, followed by several rounds that implement the subsequent linear attention layers.

---

**The $(L, a_1, \cdots, a_L)$-Hybrid Model**

**Settings.** The model operates over $L$ epochs with $L + 2$ players, indexed as $[-1 : L]$.
**Input.** Player $i \in [L]$ receives $z_i$ (from Definition 2.5), encoded in $m_{(i)} = N_{i-1}$ tokens. Player 0 and player $-1$ receive $z_0$ and $w$, respectively, each encoded in a single token (i.e., $m_{(-1)} = m_{(0)} = 1$).

---

**Communication.** For $\ell \in [0 : L]$, let $X_i^{(\ell)}$ denote the message collected by player $i \in [-1 : L]$ after $\ell$ epochs. For $\ell \in [L]$, the $\ell$-th epoch of communication proceeds as follows. For player $i \in [-1 : L]$,

- The player $i$ sends its information $X_i^{(\ell-1)}$ to all players $[i+1 : L]$.

- Each player $j \in [i+1 : L]$, based on its own information $X_j^{(\ell-1)}$ and the information $X_i^{(\ell-1)}$ from player $i$, sends a message $\Pi_{j,i}^{(\ell)}$ (termed a "soft transcript", corresponding to the full attention layer) to player $i$. The length of the message satisfies

$$|\Pi_{j,i}^{(\ell)}| = 2Hdp \cdot m_{(i)}.$$

- The player $i$ intermediately updates its collection of information as

$$X_i^{(\ell),0} := X_i^{(\ell-1)} \cup \bigcup_{j>i} \Pi_{j,i}^{(\ell)}.$$

- The players then engage in $a_\ell$ rounds of communication (corresponding to the linear attention layers). For each round $m \in [0 : a_\ell - 1]$, player $L$ sends a message $\Sigma_L^{(\ell,m)}$ of $Hd(d+1)p$ bits. Sequentially, each player $i = L - 1, \cdots, 1, 0$ receives the message $\Sigma_{i+1}^{(\ell,m)}$ (we call them "linear transcripts", corresponding to the linear attention layer) from player $i + 1$, and updates its information as

$$X_i^{(\ell),m+1} := X_i^{(\ell),m} \cup \Sigma_{i+1}^{(\ell),m},$$

then sends a message $\Sigma_i^{(\ell,m)}$ of $Hd(d+1)p$ bits to player $i - 1$. Finally, player $-1$ updates its information as

$$X_{-1}^{(\ell),m+1} := X_{-1}^{(\ell),m} \cup \Sigma_0^{(\ell),m}.$$

After $a_\ell$ linear rounds, the epoch concludes by setting $X_i^{(\ell)} = X_i^{(\ell),a_\ell}$ for all $i \in [-1, L]$.

**Output.** After the $L$-th round, the player $-1$ produces the final output based on $X_{-1}^{(L)}$.

---

We remark that the players are forgetful, the player $j$ does not remember anything sent from previous players.

# 4. Hybrid Communication Lower Bound

We demonstrate a limitation of hybrid Transformer architectures combining full attention with efficient linear attention layers on solving deep sequential composition tasks. Our main theorem shows that even abundant linear attention cannot substitute for the expressive power of full attention on certain hierarchical tasks. More precisely, we prove Theorem 2.6 via a communication complexity argument. The core of our proof is an inductive construction of indistinguishable input sets that become impossible for the model to distinguish, despite requiring different outputs.

We analyze the hybrid Transformer's computation through the lens of information flow between layers. Each full attention layer enables parallel aggregation of information from all previous positions, while linear attention layers only permit sequential, recurrent propagation. The latter cannot create the rich interactions needed for deep composition.

The $L$-sequential function composition task is specifically designed to control the contribution of the linear attention layers to the transcript space and to ensure technical requirements. Concrete choices of task parameters are given in Appendix B. The proof technique provides a general framework for analyzing hierarchical computation in structured transformers. It demonstrates that linear attention—despite its efficiency—lacks the expressive power required for deep compositional reasoning, even when augmented with limited full attention. The rest of this section provides a sketch of our proof; the detailed proof is given in Appendix B.

## 4.1. Parameters and strateggy

To prove Theorem 2.6, we construct parameters of the task such that the input length satisfies $n \leq (Hdp)^{4 \cdot 16^L}$, yet the task cannot be solved by an $(L-1, 2^{3L^2}, \cdots, 2^{3L^2})$-hybrid Transformer. We assume that $Hdp \geq 2$.

Indeed, we prove a stronger result: even if we allow pre-treatment by a single-layer Transformer, the $L$-sequential function composition task cannot be solved by a small $(L-1, 2^{3L^2}, \cdots, 2^{3L^2})$-hybrid Transformer. Equivalently, we prove that a small $(L, 0, 2^{3L^2}, \cdots, 2^{3L^2})$-hybrid Transformer cannot solve $L$-sequential function composition.

**Notation.** For notational convenience, we use $z_{-1}$ and $w$ interchangeably to denote player $-1$'s input. In the following, we elaborate on several key definitions that will be crucial to our proof.

- (Soft transcript $\Pi_{j,i}^{(\ell)}$) For any $i \in [-1 : L-1]$, $j \in [i+1 : L]$, $\ell \in [L]$, recall that $\Pi_{j,i}^{(\ell)}$ denotes the soft transcript sent from player $j$ to player $i$ at the $\ell$-th epoch of communication. Its value is determined by the inputs of players $[i : L]$ (i.e.,

$z_L, \ldots, z_i$) and is independent of the the inputs of players $[-1 : i-1]$ (i.e., $z_{i-1}, \ldots, z_{-1}$). For any fixed inputs $\widetilde{z}_L \in [N_{L-1}]^{N_{L-1}}, \ldots, \widetilde{z}_i \in [N_{i-1}]^{N_{i-1}}$, let $\Pi_{j,i}^{(\ell)}(\widetilde{z}_L, \ldots, \widetilde{z}_i)$ denote the soft transcript when player $t$ receives input $z_t = \widetilde{z}_t$ ($t \in [i : L]$).

- (Linear transcript $\Sigma_{i+1}^{(\ell),m}$) For any $i \in [-1, L-1]$, $l \in [L]$, and $m \in [0, a_\ell - 1]$, recall $\Sigma_{i+1}^{(\ell),m}$ is the linear transcript sent from player $i+1$ to player $i$ in the $(m+1)$-th linear round of the $\ell$-th epoch of communication. Its value is determined by the inputs of players $[i+1 : L]$ (i.e., $z_L, \ldots, z_{i+1}$) and is independent of the the inputs of players $[-1 : i]$ (i.e., $z_i, \ldots, z_{-1}$). For any fixed inputs $\widetilde{z}_L \in [N_{L-1}]^{N_{L-1}}, \ldots, \widetilde{z}_i \in [N_{i-1}]^{N_{i-1}}$, let $\Sigma_{i+1}^{(\ell),m}(\widetilde{z}_L, \ldots, \widetilde{z}_i)$ denote the transcript when player $t$ receives input $z_t = \widetilde{z}_t$ ($t \in [i : L]$).

- (The partial composition value) For any $\ell \in [0 : L]$, the value of $i_\ell$ is determined by $w, z_0, \ldots, z_\ell$. We write $i_\ell(\widetilde{w}, \widetilde{z}_0, \ldots, \widetilde{z}_\ell)$ to denote its value when $w = \widetilde{w}, z_0 = \widetilde{z}_0, \ldots, z_\ell = \widetilde{z}_\ell$.

**Indistinguishable decomposition.** Our key concept for the proof is *indistinguishable decomposition* introduced in (Chen et al., 2025). A indistinguishable decomposition is formed by two sets $R_{\geq \ell}$ and $Z_{<\ell}$, where $R_{\geq \ell}$ is a set of input assignments to players $[\ell : L]$ and $Z_{<\ell}$ is a set of input assignments to players $[-1 : \ell - 1]$). The key property is that for any fixed input $z_{<\ell} \in Z_{<\ell}$ for the first $\ell$ players, all assignments to $R_{\geq \ell}$ are indistinguishable to players $[-1 : \ell - 1]$ on inputs $z_{<\ell}$ after $\ell$ epochs, because they produce identical communication transcripts. Formally, we adapt the definition in (Chen et al., 2025) as follows.

**Definition 4.1** (Indistinguishable decomposition). Let $\ell \in [2 : L]$, an indistinguishable decomposition is a pair of sets $R_{\geq \ell} \subseteq A_L \times A_{L-1} \times \cdots \times A_\ell$ and $Z_{<\ell} = Z_{-1} \times \cdots \times Z_{\ell-1}$ with $Z_{-1} = A_{-1}, Z_0 \subseteq A_0, \cdots, Z_{\ell-1} \subseteq A_{\ell-1}$, such that for every $\widetilde{z}_{<\ell} \in Z_{<\ell}$, and for every $\widetilde{\alpha}_{\geq \ell}, \widetilde{\beta}_{\geq \ell} \in R_{\geq \ell}$, it satisfies:

$$\Pi_{j,i}^{(\ell')}(\widetilde{z}_{<\ell}, \widetilde{\alpha}_{\geq \ell}) = \Pi_{j,i}^{(\ell')}(\widetilde{z}_{<\ell}, \widetilde{\beta}_{\geq \ell})$$

for every $j \in [\ell : L]$, $i \in [-1 : \ell - 1]$, and $\ell' \in [\ell]$, and

$$\Sigma_{i+1}^{(\ell'),m}(\widetilde{z}_{<\ell}, \widetilde{\alpha}_{\geq \ell}) = \Sigma_{i+1}^{(\ell'),m}(\widetilde{z}_{<\ell}, \widetilde{\beta}_{\geq \ell})$$

for every $i \in [-1 : \ell - 1]$, $\ell' \in [\ell]$, and $m \in [0, a_{\ell'} - 1]$.

The utility of an indistinguishable decomposition becomes clear when $\ell = L$. In this case, for every input assignment from $Z_{<L}$ to players $[-1 : L-1]$, player $-1$ (the final output player) observes identical communication transcripts after $L$ epochs (i.e., at the end of the protocol) regardless of which input $\vec{z}_L \in R_{\geq L}$ is assigned to player $L$. Consequently, for every $\widetilde{z}_{<L} \in Z_{<L}$, the output of the protocol $L$-FuncComp($\widetilde{z}_{<L}, \widetilde{z}_L$) must be the same for every

$\widetilde{z}_L \in R_{\geq L}$. We will carefully define the set $R_{\geq \ell}$ and $Z_{<\ell}$ so that satisfying this requirement leads to a contradiction, thereby establishing the desired lower bound.

For a subset $Z_{<\ell}$, we define $\mathcal{I}_{\ell-1}(Z_{<\ell})$ to be the set of all possible values for the intermediate composition after the $(\ell-1)$-th epoch when the inputs to players $[-1 : \ell-1]$ are restricted to $Z_{<\ell}$:

$$\{i_{\ell-1}(\widetilde{z}_{-1}, \widetilde{z}_0, \ldots, \widetilde{z}_{\ell-1}) : \widetilde{z}_{-1}, \widetilde{z}_0, \ldots, \widetilde{z}_{\ell-1}) \in Z_{<\ell}\}.$$

The following lemma, as in (Chen et al., 2025), shows that the desired lower bound follows from a good enough indistinguishable configuration for $\ell = L$.

**Lemma 4.2** (Lemma B.5). *An $L$-epoch hybrid communication protocol does not solve $L$-FuncComp if there is an indistinguishable decomposition $R_{\geq L}$ and $Z_{<L}$, such that both $|R_{\geq L}|$ and $|\mathcal{I}_{L-1}(Z_{<L})|$ are large.*

Since all $z_L \in R_{\geq L}$ are indistinguishable to player $-1$ (the output player), the protocol must output the same answer for every $z_L \in R_{\geq L}$. However, because $\mathcal{I}_{L-1}(Z_{<L})$ is large, distinct $z_L \in R_{\geq L}$ would require different outputs for some input in $Z_{<L}$. This contradicts the correctness of the protocol, thereby completing the proof.

The proof of Theorem 2.6 is then completed by constructing the required decomposition via an inductive argument.

**Lemma 4.3** (Lemma B.7). *For any $\ell \in [2 : L]$, we can construct by induction an indistinguishable decomposition $(R_{\geq \ell}, Z_{<\ell})$, where $R_{\geq \ell} \subseteq A_L \times A_{L-1} \times \cdots \times A_\ell$ and $Z_{<\ell} = Z_{-1} \times Z_0 \times \cdots \times Z_{\ell-1}$, with $Z_{-1} = [n_1 \cdots n_{L-1}]$, $Z_0 \subseteq A_0$ and $Z_i \subseteq A_i$, together with the soft transcript from players $[\ell : L]$ to $[-1 : \ell-1]$ for the first $\ell$ epochs, when the players $[-1 : \ell-1]$ take input from $Z_{<\ell}$ and the linear transcript from player $\ell$ to $\ell-1$ for the first $\ell$ epochs, when the players $[-1 : \ell-1]$ take input from $Z_{<\ell}$, such that the soft transcripts and linear transcripts are consistent, and that $|R_{\geq \ell}|$ and $|\mathcal{I}_{\ell-1}(Z_{<\ell})|$ are large.*

### 4.2. The Initial step

We first consider the base case $\ell = 2$.

**Step 1: Choosing $Z_0, Z_1$.** We set $Z_0 = [x_0]$. The next step is to select the set $Z_1 \subseteq A_1$. Consider all possible first epoch messages from the player $1$ to the player $-1$, The total number of of distinct such message patterns is $2^{2Hdp \cdot |Z_{-1}|} = 2^{2Hdp(n_1 \cdots n_{L-1})}$. We select by the pigeonhole principle a message pattern $\widetilde{\Psi}_{1,-1}^{(1)} \in \{0,1\}^{2Hdp \cdot (n_1 \cdots n_{L-1})}$ such that the consistency set $S$ of input $\widetilde{z}_1 \in A_1$ satisfies $|S| \geq |A_1| \cdot 2^{-2Hdp \cdot (n_1 \cdots n_{L-1})}$. We then retract a suitable subset $Z_1 \subseteq S$ by the following lemma.

**Lemma 4.4** (Lemma B.8). *There exists a subset $Z_1 \subseteq S$ such that $\mathcal{I}_1(Z_{<2})$ is large.*

The next step is to fix the transcripts from players $j \in [2 : L-1]$ to players $i = -1, 0, 1$ at the first two epochs.

**Step 2.1: Fixing the transcripts to player $-1$.** We begin with the first epoch. Our goal is to fix soft transcripts sent to player $-1$ in the first epoch for every $\widetilde{z}_1 \in Z_1, \widetilde{z}_0 \in Z_0, \widetilde{z}_{-1} \in Z_{-1}$. The first epoch message from player $j$ to player $-1$ depends only on $\widetilde{z}_{-1}$ and player $j$'s input, and is independent of $\widetilde{z}_1, \widetilde{z}_0$. Therefore, the total number of possible such transcript patterns is at most $2^{2Hdp \cdot |Z_{-1}|} = 2^{2Hdp \cdot (n_1 \cdots n_{L-1})}$. By the pigeonhole principle, we can choose a fixed pattern with the consistency set of maximal size $|C_1| \geq |A_L| \cdots |A_2| \cdot 2^{-2Hdp \cdot (n_1 \cdots n_{L-1}) \cdot L}$.

For the second epoch, the goal is to fix soft transcripts sent to player $-1$ in the second epoch for every $\widetilde{z}_1 \in Z_1, \widetilde{z}_0 \in Z_0, \widetilde{z}_{-1} \in Z_{-1}$. Crucially, the transcript from player $j$ depends only on the information state $X_{-1}^{(1)}$ and $X_j^{(1)}$, which are themselves independent of the choice of $z_1 \in Z_1$. This independence holds because the only component of $X_{-1}^{(1)}$ and $X_j^{(1)}$ that depends on $z_1$ is the first epoch message from player $1$ to $-1$, which is fixed. We choose transcripts with the consistency set $C_2$ of maximal size.

**Step 2.2: Fixing the transcripts to player $0$ and player $1$.** Similarly, We fix the value of soft transcripts sent to player $0$ in the first two epochs so that the consistency set $C_3$ has maximal size.

We follow the same strategy to fix the value of soft transcripts and the linear transcripts sent to player $1$ in the first two epochs so that the consistency set $C_4$ attains its maximal size. Take $R_{\geq 2}$ to be $C_4$, this concludes the proof for the base case $\ell = 2$.

### 4.3. Inductive step

Suppose we are done up to $\ell \in [2 : L-1]$. We continue our construction for $\ell + 1$.

The key insight is that $Z_\ell$ is indistinguishable to players $[-1 : \ell-1]$ after $\ell$ epochs, when these players take input from $Z_{<\ell}$. Hence, the $(\ell+1)$-th epoch transcripts to players $[-1 : \ell-1]$ are independent of the choice of $z_\ell \in Z_\ell$.

**Step 1: Choosing the set $Z_\ell$.** Recall that the size of $R_{\geq \ell}$ is large. We would like to select a rectangular subset from $R_{\geq \ell}$. As in (Chen et al., 2025), we have the following lemma.

**Lemma 4.5** (Lemma B.9). *There exists a subset $S^{(\ell)} = S_1^{(\ell)} \times S_2^{(\ell)}$, where $S_1^{(\ell)} \subseteq A_L \times \cdots \times A_{\ell+1}$, $S_2^{(\ell)} \subseteq A_\ell$, such that $S_1^{(\ell)}$ and $\mathcal{I}_{\ell-1}(S_2^{(\ell)})$ are large.*

We take $Z_\ell = S_2^{(\ell)}$ and $S_{\geq \ell+1} = S_1^{(\ell)}$. Next, we are going to fix the transcripts. Recall that we need to fix all transcripts from players $[\ell+1 : L]$ to players $[-1 : \ell]$ in the first

$\ell + 1$ epochs, when players $[-1 : \ell]$ receive input from $Z_{\leq \ell} = Z_{-1} \times Z_0 \times \cdots \times Z_\ell$. We proceed in a few steps.

**Step 2.1: Fixing the transcripts to players $[-1 : \ell - 1]$ in the first $\ell$ epochs.** We simply the transcripts given by the induction hypothesis. Lemma B.10 guaranties the consistency.

**Step 2.2: Fixing the transcripts to players $[-1 : \ell - 1]$ at the $(\ell + 1)$-th epoch.** Our key insight is that $Z_\ell$ is indistinguishable to players $[-1 : \ell - 1]$ when they take input from $Z_{\leq \ell - 1}$. Hence, their transcripts are independent of the choice of $z_\ell \in Z_\ell$. We consider the soft transcripts from players $[\ell + 1, L]$ to players $[-1 : \ell - 1]$ at the $(\ell + 1)$-th epoch

$$\Phi^{(\ell+1)} = \left( \Phi_{j,i}^{(\ell+1)} \right)_{j \in [\ell+1:L], i \in [-1:\ell-1]}$$

where

$$\Phi_{j,i}^{(\ell+1)} = \left( \Phi_{j,i}^{(\ell+1)}(\widetilde{z}_{\ell-1}, \ldots, \widetilde{z}_i) \right)_{\widetilde{z}_{\ell-1} \in Z_\ell \ldots, \widetilde{z}_i \in Z_i}$$

$$\text{and} \quad \Phi_{j,i}^{(\ell+1)}(\widetilde{z}_{\ell-1}, \ldots, \widetilde{z}_i) \in \text{domain}(\Pi_{j,i}^{(\ell+1)})$$

Comparing $\Lambda_{j,i}^{(\ell+1,\ell+1)}$ with $\Phi_{j,i}^{(\ell+1)}$, we note that $\Phi_{j,i}^{(\ell+1)}$ is independent of $\widetilde{z}_\ell \in Z_\ell$. For any $\Phi^{(\ell+1)}$, define $S(\Phi^{(\ell+1)})$ to be the set of all $(\widetilde{z}_L, \ldots, \widetilde{z}_{\ell+1}) \in S_{\geq \ell+1}$ that are consistent with the transcripts $\Phi^{(\ell+1)}$.

**Lemma 4.6** (Lemma B.11). *We have*

$$\bigcup_{\Phi^{(\ell+1)}} S(\Phi^{(\ell+1)}) = S_{\geq \ell+1}.$$

Now as in (Chen et al., 2025), we take the transcripts with maximal consistency set $T_{\geq \ell+1} = S(\widetilde{\Phi}^{(\ell+1)}) \subseteq S_{\geq \ell+1}$.

**Step 2.3: Fixing the transcript to player $\ell$.** This follows a greedy selection strategy. First, consider the soft transcripts sent to player $\ell$ in the first $\ell + 1$ epochs

$$\Psi = \left( \Psi_{j,\ell}^{(\ell')}(\widetilde{z}_\ell) \right)_{j \in [\ell+1:L], \ell' \in [\ell+1], \widetilde{z}_\ell \in Z_\ell}$$

$$\text{where} \quad \Psi_{j,\ell}^{(\ell')}(\widetilde{z}_\ell) \in \text{domain}(\Pi_{j,\ell}^{(\ell')})$$

We can upper-bound the number of distinct $\Psi$ and then lower-bound the size of the maximal consistency set $T(\Psi)$ via pigeonhole principle (Lemma B.13).

Next, we fix linear transcripts to player $\ell$ in the first $\ell + 1$ epochs to maintain the maximal consistency set. We can then take $R_{\geq \ell+1}$ to be this consistency set, and this completes the induction step.

## 5. Lower bound for sparse attention

We now prove Theorem 2.8. For convenience, we assume that $B$ divides $n$. The proof proceeds by relating a $(B, k)$-sparse attention to the following communication model.

---

**Model for $2$-Sum via sparse attention**

**Settings.** There are $\frac{n}{B} + 1$ players.
**Input.** Each of the first $\frac{n}{B}$ players receives $B$ tokens. The last player receives a single token.
**Communication.** Each of the first $\frac{n}{B}$ players sends a $Hdp$-bit message to the last player. Subsequently, the last player may select $k$ of these players and access their full information.
**Output.** The last player produces an answer based on its gathered information.

---

In this model, each block of $B$ tokens corresponds to one of the first $\frac{n}{B}$ players, the last token corresponds to the final player, and the compressed representation (of size $Hdp$ bits) corresponds to the message sent to the final player.

One can prove that, for the last player to output the correct answer, each of the first players must communicate the set of its $B$ tokens via a compressed message of $Hdp$ bits. This implies the desired result; details are given in Appendix A.6.

## 6. Conclusion

In this work, we establish a unified hierarchy of expressive power for efficient attention mechanisms within a communication complexity framework. By analyzing hybrid architectures, we prove an unconditional lower bound: even an abundance of linear attention layers cannot substitute for the compositional power of a single full attention layer in solving deep sequential function composition tasks.

These tasks formally model the requirement for multi-step sequential computation within a single forward pass, capturing problems like multi-hop retrieval where each step's output defines the input context for the next. We discuss the choices of the tasks in Appendix E.

Our theoretical framework introduces carefully constrained "retrieval scopes" at each internal computational step to enable rigorous analysis. This result provides a theoretical justification for the empirical observation that simply interleaving linear and full attention yields limited gains on deep reasoning problems.

Furthermore, we identify a fundamental limitation of single-layer sparse attention mechanisms based on block compression and selection. For tasks like 2-Sum that require uniform pair-wise comparisons, any such sparse mechanism is provably weaker than full attention unless its effective capacity scales with the block size.

A discussion on the limitations of our framework can be found in Appendix E.

Collectively, our results offer a principled theoretical framework for understanding efficient attention variants. They

suggest that future architectures for long-context processing must either embrace the expressive power of full attention through smarter approximations, or explicitly design around these limitations—for instance, by employing task-sufficient sparsity patterns or incorporating mechanisms to bypass the communication bottlenecks we have identified. The communication models and proof techniques introduced here may serve as a foundation for further theoretical analysis of structured Transformers.

## Impact Statement

This paper presents work whose goal is to advance the field of Machine Learning. There are many potential societal consequences of our work, none which we feel must be specifically highlighted here.

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

# A. Lower bounds for linear attention, log-linear attention and sparse attention

In this section, we present communication models for different attention mechanisms on various tasks. We abstract the process of solving such tasks via linear attention and log-linear attention as certain communication protocols, and prove lower bounds in Theorems A.3, A.6, A.5, A.8 and A.9 via communication complexity arguments.

## A.1. Definitions

### A.1.1. LOG-LINEAR ATTENTION

In the log-linear attention mechanism proposed in (Guo et al., 2025), the output of the $\ell$-th attention layer becomes

$$y_i^{(\ell,h)} = \sum_{r=0}^{R-1} \lambda_i^{(r,\ell,h)} S_i^{(r,\ell,h)} Q x_i^{(\ell-1,h)} \tag{3}$$

where $\lambda_i^{(r,\ell,h)}$ are weights that depend only on $x_i^{(\ell-1,h)}$, $R = \lceil \log_2 i + 1 \rceil + 1$ and $S_i^{(r,\ell,h)}$ are hidden states. In (Guo et al., 2025), these states are calculated via the recursion

$$S_i^{(r,\ell,h)} = \begin{cases} V x_i^{(\ell-1,h)} (K x_i^{(\ell-1,h)})^\top & \text{if } r=0 \\ 0 & \text{if } 0<r\leq \text{lssb}(i) \\ \sum_{r'=0}^{r-1} S_{i-1}^{(r',\ell,h)} & \text{if } r=\text{lssb}(i)+1 \\ S_{i-1}^{(r,\ell,h)} & \text{if } r>\text{lssb}(i)+1 \end{cases},$$

where $\text{lssb}(t) = \max\{\ell \in \mathbb{N} \mid 2^\ell \text{ divides } t\}$. Under such settings, a single-layer one-head log-linear attention of dimension $d$ can be viewed as an RNN of hidden dimension $3d^2$, with hidden states comprising the 3 values of $S_i^{(r)}$ that are potentially non-zero, that is, $h_i = (S_i^{(0)}, S_i^{(\text{lssb}(i))}, S_i^{(\text{lssb}(i)+1)})$.

We consider a more general setting

$$S_i^{(r,\ell,h)} = f^{(r,\ell,h)}\left((S_{i-1}^{(r',\ell,h)})_{r'\in[0,r-1]}, x_i^{(\ell-1,h)}\right) \tag{4}$$

where $f^{(r,\ell,h)}$ are certain pre-trained functions. This enables computation with logarithmic time and memory, as it only requires maintaining a set of hidden states whose size is logarithmic in the sequence length.

### A.1.2. FUNCTION EVALUATION AND PERMUTATION COMPOSITION

**Definition A.1** (Function evaluation). In a function evaluation task $\mathsf{Eva}(f, x)$, the input consist of a function $f : [n] \to [n]$ represented by $n$ tokens encoding $f(1), \cdots, f(n)$, and an element $x \in [n]$ described by a single token, and the goal is to output $\mathsf{Eva}(f, x) = f(x) \in [n]$.

The permutation composition task can be viewed as $n$ parallel instances of the function evaluation task.

**Definition A.2** (Permutation composition). In a permutation composition task $\mathsf{PerCom}(\sigma, \tau)$, the input consist of two bijections $\sigma, \tau : [n] \to [n]$ (each occupies $n$ tokens to describe $\sigma(1), \cdots, \sigma(n)$ and then $\tau(1), \cdots, \tau(n)$), and the goal is to output their composition in the form of the sequence $\sigma(\tau(1)), \cdots, \sigma(\tau(n))$.

## A.2. Lower bound for linear attention on $\mathsf{Eva}$

In this subsection, we establish communication models of linear attention and log-linear attention for different tasks.

**Theorem A.3.** An $L$-layer linear attention cannot solve $\mathsf{Eva}$ whenever $LHd(d+1)p < n \log n$, while a single-layer full attention solves the task with $Hdp = O(\text{poly} \log n)$. The same holds for linear attention with CoT.

Theorem A.3 can be implied by the following result since linear attention can be viewed as an RNN (Lemma 2.2).

**Theorem A.4.** An $L$-layer RNN with $H$ heads, hidden dimension $m$ and precision $p$ cannot solve $\mathsf{Eva}$ whenever $LHmp < n \log n$. The same holds for RNNs with CoT.

To prove this theorem, we first present the communication model for an RNN without CoT to solve $\mathsf{Eva}$.

---

**Communication Model for** Eva **via RNN**

**Settings.** There are 2 players: Alice and Bob. They communicate in $L$ rounds. A third player, Charles, is included if CoT is enabled.

**Input.** Alice receives $n$ tokens as input (corresponding to $f(1), \cdots, f(n)$), Bob receives one token as input (corresponding to $x$), and Charles processes the tokens generated during the CoT steps.

**Communication.** For $\ell \in [L]$, during the $\ell$-th round of communication, Alice sends a message $M_\ell$ of $Hmp$ bits to Bob. Bob then sends a message $\tilde{M}_\ell$ of $H(m+d)p$ bits to Charles if CoT is enabled.

**Output.** At the end of the $L$-th round, Charles outputs a message based on its information if CoT is enabled; otherwise, Bob outputs a message based on its information.

---

The $L$ communication rounds correspond to the $L$ layers of the RNN. In each round $\ell$, the message $M_\ell$ that Alice sends to Bob corresponds to the hidden state $h_n^{(\ell,h)}$ at position $n$ of the $\ell$-th layer and $h$-th head. Bob can then use this information to update the hidden state $h_{n+1}^{(\ell,h)}$ and then compute $y_{n+1}^{(\ell,h)}$, the output of the $\ell$-layer at position $n+1$.

In the variant of this model describing the computation of function evaluation via RNN with CoT, we introduce a third participant, Charles, to model the computation involving the additional tokens generated during Chain-of-Thought (CoT) steps. In this extended model, Bob must send both the hidden state and the output $y_{n+1}^{(\ell,h)}$ to Charles. This output $y_{n+1}^{(\ell,h)}$ serves as the initial value for the first token that Charles processes. Consequently, the size of the message (in bits) becomes $Hmp + Hdp$.

The key insight is that Charles's knowledge is entirely derived from the information sent by Bob. Therefore, if Charles can compute the final output, then Bob, who possesses at least the same information, must also be able to compute it.

The key to the proof is that the total number of communicated bits is $LHmp$, which should be at least $\log(n^n) = n \log n$ for Bob to distinguish all the functions.

We also establish communication models for PerCom and 2-Sum, detailed proofs are given in Appendix A.

*Proof of Theorem A.4.* We first consider the case without CoT. Over $L$ rounds, Alice sends a total of $LHmp$ bits to Bob, while Bob sends no messages to Alice. These messages allow Bob to distinguish at most $2^{LHmp}$ different functions.

If $2^{LHmp} < n^n$, or equivalently, if $LHmp < n \log n$, then there exist two distinct functions $f_1, f_2 : [n] \to [n]$ ($f_1 \neq f_2$) that are indistinguishable to Bob; that is, the sequence of messages from Alice is identical for $f_1$ and $f_2$. Since $f_1 \neq f_2$, there exists some element $x \in [n]$ such that $f_1(x) \neq f_2(x)$. If Bob's input is $x$, it becomes impossible to output the correct answer, as Bob cannot determine whether to output $f_1(x)$ or $f_2(x)$.

In the CoT case, all of Charles' information is derived from Bob. Therefore, if Charles can determine the answer $f(x)$, then Bob must also be able to compute it. This contradicts the lower bound established for the case without CoT. $\qquad\square$

### A.3. Lower bound for log-linear attention on Eva

In this subsection, we establish the communication model for log-linear attention to solve Eva, yielding a lower bound.

**Theorem A.5.** *An $L$-layer log-linear attention cannot solve* Eva *whenever $LHd^2p < \Theta(n)$, while a single-layer full attention solves the task with $Hdp = O(\text{poly} \log n)$. The same holds for log-linear attention with CoT.*

Recall from (3) and (4) that to compute the output $y_{n+1}^{\ell,h}$, Bob requires the hidden states $S_{n+1}^{(r,\ell,h)}$ for all $r \in [0, \lceil \log(n+1)+1 \rceil]$. These states, in turn, depend on the previous hidden states $S_n^{(r,\ell,h)}$, for $r \in [0, \lceil \log(n)+1 \rceil]$. Therefore, Alice only needs to send Bob these $O(\log n)$ hidden states. This requires $O(d^2p \log n)$ bits per head, resulting in a total message size of $O(Hd^2p \log n)$ bits per layer.

---

**The Communication Model for** Eva **via log-linear attention**

**Settings.** Two players, Alice and Bob, communicate in $L$ rounds. A third player, Charles, is included if CoT is enabled.

**Input.** Alice receives $n$ tokens as input, and Bob receives one token as input. Charles receives the tokens

---

corresponding to the CoT steps.

**Communication.** For $\ell \in [L]$, during the $\ell$-th round, Alice sends a message $M_\ell$ of $O(Hd^2 p \log n)$ bits to Bob. If CoT is enabled, Bob then sends a message $\tilde{M}_\ell$ of $O(Hd^2 p \log(n+1) + Hdp)$ bits to Charles.

**Output.** At the end of the $L$-th round, Charles produces an output based on all received information if CoT is enabled; otherwise, Bob outputs a message based on its information.

In the model for log-linear attention with CoT, we introduce a new player, Charles, to deal with the tokens of CoT steps. In this model, Bob sends Charles the output $y_{n+1}^{(\ell,h)}$ along with the relevant hidden states, forming a message of $Hd^2 p \log(n+1) + Hdp$ bits. However, as in the proof of Theorem A.3, Charles's entire information state is derived from Bob. Therefore, introducing Charles does not alter the lower bound established for the case without CoT.

Now we prove the lower bound for log-linear attention on Eva.

*Proof of Theorem A.5.* For the case without CoT, during the $L$-round communication process, Alice sends Bob a total of $O(LHd^2 p \log n)$ bits, while Bob does not send any message to Alice. This communication can distinguish at most $N = 2^{O(LHd^2 p \log n)}$ different objects.

If $N < n^n$, or equivalently, if $LHd^2 p < \Theta(n)$, there should be two distinct functions $f_1, f_2 : [n] \to [n]$ that are indistinguishable for Bob (that is, the messages that Alice sends to Bob when the input is $f_1$ and such messages when the input is $f_2$ coincide). Since $f_1 \neq f_2$, there exists an element $x \in [n]$ such that $f_1(x) \neq f_2(x)$. If Bob's input is $x$, then it is impossible to produce the correct output, as Bob cannot determine whether the answer should be $f_1(x)$ or $f_2(x)$.

In the CoT case, Charles's entire knowledge is derived from Bob. Therefore, if Charles can compute $f(x)$, then Bob must also be able to compute it. This contradicts the lower bound established for the non-CoT case. □

### A.4. Lower bound for linear attention and log-linear attention on PerCom

Analogous to the previous results, we establish communication models for RNNs and log-linear attention to solve PerCom.

**Theorem A.6.** *An $L$-layer linear attention cannot solve* PerCom *whenever $LHd(d+1)p < \log(n!) = \Theta(n \log n)$, while a single-layer full attention solves it with $Hdp = O(\text{poly} \log n)$. The same holds for linear attention with CoT.*

For RNN, we prove the following theorem, which, combining with Lemma 2.2 and the construction in Section C, implies Theorem A.6.

**Theorem A.7.** *An $L$-layer RNN with $H$ heads, hidden dimension $m$ and precision $p$ cannot solve* PerCom *whenever $LHmp < \log(n!)$. The same result holds for linear attention with CoT.*

**The Communication Model for PerCom via RNN**

**Settings.** Two players, Alice and Bob, communicate over $L$ rounds. A third player, Charles, is included if CoT is enabled.

**Input.** Alice receives $n$ tokens as input (corresponding to $\sigma(1), \cdots, \sigma(n)$) and Bob receives $n$ tokens as input (corresponding to $\tau(1), \cdots, \tau(n)$). If CoT is allowed, Charles receives the tokens corresponding to the CoT steps.

**Communication.** For $\ell \in [L]$, during the $\ell$-th round, Alice sends a message of $Hmp$ bits to Bob. If CoT is allowed, Bob then sends a message of $H(m+d)p$ bits to Charles.

**Output.** At the end of the $L$-th round, the output is produced by Charles if CoT is enabled; otherwise, it is produced by Bob.

*Proof of Theorem A.7.* Similarly, during the $L$-round communication process, Alice sends a total of $LHmp$ bits, while Bob does not send any message to Alice. For Bob to distinguish all possible permutations $\sigma$, one should have $2^{LHmp} \geq n!$. Applying Stirling's formula (Stirling, 1753)

$$n! \sim \sqrt{2\pi n} \left(\frac{n}{e}\right)^n \quad \text{as} \quad n \to \infty$$

we conclude that $LHmp = \Omega(n \log n)$ is necessary for the RNN to solve PerCom. For the CoT case, the same argument as in the proof of Theorem A.4 proves the desired result. □

**Theorem A.8.** *An $L$-layer log-linear attention cannot solve* PerCom *whenever $LHd^2p < \frac{\log(n!)}{\log n} = \Theta(n)$, while a single-layer full attention solves the task with $Hdp = O(\text{poly} \log n)$. The same holds for log-linear attention with CoT.*

*Proof of Theorem A.8.* We consider the following communication model:

---

**The Communication Model for** PerCom **via log-linear attention**

**Settings.** Two players, Alice and Bob, communicate over $L$ rounds. A third player, Charles, is included if CoT is enabled.

**Input.** Alice receives $n$ tokens as input, and Bob receives $n$ token. If CoT is allowed, Charles receives the tokens corresponding to the CoT steps.

**Communication.** For $\ell \in [L]$, during the $\ell$-th round, Alice sends a message of $O(Hd^2p \log n)$ bits to Bob. If CoT is allowed, Bob then sends a message of $O(Hd^2p \log(2n) + Hdp)$ bits to Charles.

**Output.** At the end of the $L$-th round, Charles outputs a message based on its information if CoT is allowed, otherwise Bob outputs a message based on its information.

---

For a correct output, Bob must identify the permutation $\sigma$. To distinguish all possibilities, one must have $2^{O(Hd^2p \log n)} \geq n!$. This implies $Hd^2p \geq \Omega\left(\frac{\log(n!)}{\log n}\right) = \Theta(n)$ as in the proof of Theorem A.7. $\square$

## A.5. Lower bound for linear attention and log-linear attention on $2$-Sum

**Theorem A.9.** *An $L$-layer linear attention cannot solve $2$-Sum whenever $LHd(d+1)p < \Theta(n \log n)$, and an $L$-layer log-linear attention cannot solve $2$-Sum whenever $LHd^2p < \Theta(n)$, while a single-layer full attention solves the task with $Hdp = O(\text{poly} \log n)$. The same holds for linear attention and log-linear attention with CoT.*

*Proof of Theorem A.9.* We first establish the lower bound for linear attention. Consider the 2-Sum task with an input sequence of length $n$. We focus on the last token $x_n$ and its corresponding output $y_n$. The value of $y_n$ depends on whether there exists an index $j < n$ such that $x_j + x_n \equiv 0 \mod n$.

We take $M = n^2$ in the task. We adopt a communication model as follows.

---

**The Communication Model for** $2$-Sum **via linear attention**

**Settings.** Two players, Alice and Bob, communicate over $L$ rounds. A third player, Charles, is included if CoT is enabled.

**Input.** Alice receives the first $n-1$ tokens $(x_1, \ldots, x_{n-1})$ and Bob receives the last token $x_n$. If CoT is allowed, Charles receives the tokens corresponding to the CoT steps.

**Communication.** For $\ell \in [L]$, during the $\ell$-th round, Alice sends a message of $H \cdot (d^2 + d) \cdot p$ bits to Bob (corresponding to the hidden state of the $\ell$-th layer at position $n-1$). If CoT is allowed, Bob then sends a message of $H \cdot (d^2 + d) \cdot p + H \cdot d \cdot p$ bits to Charles (the hidden state and the output of the $\ell$-th layer for position $n$).

**Output.** At the end of the $L$-th round, Charles outputs $y_n$ based on its information if CoT is allowed; otherwise Bob outputs $y_n$ based on its information.

---

If $LHd(d+1)p < \Theta(n \log n)$ such that $2^{LHd(d+1)p} < \binom{n^2}{n-1}$, there exist two sequences of the first $n-1$ tokens, say $S_1$ and $S_2$, that yield identical hidden states but form different sets $\bar{S}_1$ and $\bar{S}_2$ of values. Therefore, there exists a value $v$ that appears in $S_1$ but not in $S_2$ (or vice versa). Without loss of generality, assume $v$ appears in $S_1$ but not in $S_2$. Set Bob's token to $x_n = -v \mod M$.

For $S_1$, we have $y_n = 1$ because there exists $j$ with $x_j = v$ such that $x_j + x_n \equiv 0 \mod n$. For $S_2$, since $v$ does not occur, no such $j$ exists, so $y_n = 0$. However, because the hidden state is the same for both sequences, the model must produce the same output, leading to an error in (at least) one case. Hence, linear attention cannot solve 2-Sum whenever $LHd(d+1)p < \Theta(n \log n)$.

For log-linear attention, we employ a similar communication model. As in the proof of Theorem A.5, the hidden state

passed from Alice to Bob has size $O(LHd^2p\log n)$ bits. If $LHd^2p\log n < \Theta(n\log n)$, i.e., $LHd^2p < \Theta(n)$, then by an analogous argument, there exist two sequences $S_1$ and $S_2$ of the first $n-1$ tokens that induce the same hidden state form different sets of values. Choosing $x_n = -v \mod n$ for a value $v$ that appears in $S_1$ but not in $S_2$ forces a contradiction, as the model must output the same $y_n$ for both sequences while the correct outputs differ. Therefore, log-linear attention cannot solve 2-Sum whenever $LHd^2p < \Theta(n)$.

---

**The Communication Model for $2$-Sum via log-linear attention**

**Settings.** Two players, Alice and Bob, communicate over $L$ rounds. A third player, Charles, is included if CoT is enabled.

**Input.** Alice receives the first $n-1$ tokens $(x_1, \ldots, x_{n-1})$ and Bob receives the last token $x_n$. If CoT is allowed, Charles receives the tokens corresponding to the CoT steps.

**Communication.** For $\ell \in [L]$, during the $\ell$-th round, Alice sends a message of $O(Hd^2p\log n)$ bits to Bob. If CoT is allowed, Bob then sends a message of $O(Hd^2p\log n) + Hdp$ bits to Charles.

**Output.** At the end of the $L$-th round, Charles outputs $y_n$ based on its information if CoT is allowed, otherwise Bob outputs $y_n$ based on its information.

---

The upper bound—that a single-layer Transformer decoder with full attention solves 2-Sum with $Hdp = O(\log n)$—follows from (Sanford et al., 2023), Theorem 6.

For the CoT variants, the same lower bounds apply, as all information Charles receives originates from Bob (If Charles could compute the output, Bob could as well, contradicting the lower bounds established without CoT). This completes the proof of Theorem A.9. □

### A.6. Lower bound for sparse attention

*Proof of Theorem 2.8.* Without loss of generality, we assume that $B$ divides $n$, and we prove that a sparse attention mechanism with small parameters cannot correctly output the expected value at position $n+1$. We define the following communication model for a $(B, k)$-sparse attention to calculate this output.

---

**The Communication Model for $2$-Sum via sparse attention**

**Settings.** There are $\frac{n}{B} + 1$ players.

**Input.** Each of the first $\frac{n}{B}$ players receives $B$ tokens, and the last player receives one token.

**Communication.** Each of the first $\frac{n}{B}$ players sends a message of $Hdp$ bits to the last player; the last player then chooses $k$ players and gets all of their information.

**Output.** The last player outputs a message based on its information.

---

In this model, each block corresponds to a player, and the compressed tokens form the messages sent to the last token.

For the last player to output the correct answer, each of the first players must communicate the set of its $B$ tokens via a compressed message of $Hdp$ bits. In fact, if this is not the case, that is, there exists two tuples $(a_i)_{i\in[B]}$ and $(b_i)_{i\in[B]}$ such that $f(a_1, \cdots, a_B) = f(b_1, \cdots, b_B)$, but $\{a_1, \cdots, a_B\} \neq \{b_1, \cdots, b_B\}$, we can consider the case that the input of each block is either $(a_i)_{i\in[B]}$ or $(b_i)_{i\in[B]}$.

In this case, all the compressed tokens would be identical, and so would the selection scores for each block. Let $n - x_{n+1}$ be an element of $\{a_1, \cdots, a_B\} \setminus \{b_1, \cdots, b_B\}$, and consider the case that all the $k$ selected blocks have input $(b_i)_{i\in[B]}$. In such case, the last player cannot distinguish between $(a_i)_{i\in[B]}$ and $(b_i)_{i\in[B]}$ for the remaining blocks. Hence, it cannot correctly output the answer.

Therefore, the compressed message of $Hdp$ bits should communicate the set of $B$ tokens, which is a subset of $[n]$ of size at most $B$, so we have

$$2^{Hdp} \geq \sum_{j \leq B} \binom{n}{j} \geq \binom{n}{B} \sim \frac{n^B}{B!},$$

this implies $Hdp = \Omega(B\log n)$ and completes the proof. □

# B. Hybrid Communication Lower Bound

In this section, we prove Theorem 2.6. We select parameters following the framework of (Chen et al., 2025) and adapt their techniques of constructing an indistinguishable decomposition to derive the result.

## B.1. Parameters and strateggy

To prove Theorem 2.6, we construct parameters of the task such that the input length satisfies $n \leq (Hdp)^{4 \cdot 16^L}$, yet the task cannot be solved by an $(L, 2^{3L^2}, \cdots, 2^{3L^2})$-hybrid Transformer. We use the following parameters throughout this section:

$$K = (HdpL)^8 \cdot 8^{2L^2}, \quad m = K^{\sum_{\ell \in [0:L-1]} 8^\ell + 1}, \tag{5}$$

$$n_\ell = K^{4 \cdot 8^{L-\ell-1}}, \forall \ell \in [L-1]. \tag{6}$$

We assume that $Hdp \geq 2$. Recall from Definition 2.5 that

$$N_\ell = m \cdot \prod_{\ell' \in [\ell]} n_{\ell'} \quad \forall \ell \in [0:L-1]. \tag{7}$$

We also define the following auxiliary parameters:

$$x_\ell = K^{8^{L-\ell-1}}, \forall \ell \in [0:L-1] \tag{8}$$

$$A_\ell = \left[ N_{\ell-1}^{N_{\ell-1}} \right], \forall \ell \in [L] \tag{9}$$

$$\Delta_\ell = 2^{4\sqrt{K}(x_0...x_{\ell-2}) \cdot (n_1...n_{L-1})}, \forall \ell \in [2:L]) \tag{10}$$

$$\Theta_\ell = 8^{-L\ell}(x_0 \ldots x_\ell) \cdot (n_1 \ldots n_{\ell-1}), \forall \ell \in [L-1]. \tag{11}$$

For notational convenience, we also define $A_{-1} = \prod_{i=1}^{L-1}[n_i]$ and $A_0 = [m]$. Recall that we denote the query $w$ by $z_{-1}$. Thus, player $i$ receives an input from the set $A_i$ for every $i \in [-1:L]$. Note that we view an element of $A_\ell = \left[ N_{\ell-1}^{N_{\ell-1}} \right]$ can be viewed as a function $[N_{\ell-1}] \to [N_{\ell-1}]$.

We first prove that the task has the desired input size.

**Lemma B.1.** *For the L-sequential function composition task with the parameters defined in (6), the input prompt length* $n = 2 + N_0 + N_1 + \cdots + N_{L-1}$ *satisfies* $n \leq (Hdp)^{4 \cdot 16^L}$.

*Proof.* From the parameter definitions in (6) and (7), it follows that $2 \leq N_0$ and $2N_{\ell-1} \leq N_\ell$ for all $\ell \in [1, L-1]$. Consequently, the total input length can be bounded as follows:

$$n = 2 + N_0 + N_1 + \cdots + N_{L-1}$$

$$(2 \leq N_0, 2N_{\ell-1} \leq N_\ell) \quad \leq 2N_{L-1} = 2m \cdot \prod_{\ell \in [L-1]} n_\ell = 2 \cdot K^{\sum_{\ell \in [0:L-1]} 8^\ell + 1 + \sum_{\ell \in [1:L-1]} 4 \cdot 8^{L-\ell-1}}$$

$$= 2 \cdot \left( (HdpL)^8 \cdot 8^{2L^2} \right)^{\frac{12}{7} \cdot 8^{L-1} + \frac{2}{7}}$$

$$(Hdp \geq 2) \quad \leq (Hdp)^{1 + \left( \frac{12}{7} \cdot 8^{L-1} + \frac{2}{7} \right)(8 \log L + 8 + 6L^2)}$$

$$\leq (Hdp)^{2 \cdot 8^{L-1}(8 \log L + 8 + 6L^2)}$$

$$(1 + \log L \leq L) \quad \leq (Hdp)^{2 \cdot 8^{L-1}(8L + 6L^2)}$$

$$(2L \leq 2^L, L^2 \leq 2^L) \quad \leq (Hdp)^{2 \cdot 8^{L-1} \cdot 10 \cdot 2^L} \leq (Hdp)^{4 \cdot 16^L} \qquad \square$$

It remains to show that the task cannot be solved by an $(L, 2^{3L^2}, \cdots, 2^{3L^2})$-hybrid Transformer. Indeed, we prove the following stronger result.

**Theorem B.2.** *No deterministic* $(L, a_1, \cdots, a_L)$-*hybrid communication protocol solves L-*FuncComp *under the following assumptions.*

1. $a_1 \leq 1$,

2. *For all $\ell \in [L-1]$, we have*

$$Hd(d+1)p(a_1 + \cdots + a_{\ell+1}) \leq \sqrt{K}(x_0 \cdots x_{\ell-1}) \cdot (n_1 \cdots n_{L-1}). \tag{12}$$

We now show that our $(L, 1, 2^{3L^2}, \cdots, 2^{3L^2})$ satisfies the assumptions of Theorem B.2. Consequently, Theorem B.2 implies Theorem 2.6, since an $(L-1, 2^{3L^2}, \cdots, 2^{3L^2})$-hybrid Transformer can be viewed as an $(L, 1, 2^{3L^2}, \cdots, 2^{3L^2})$-Transformer with trivial MLP layer $x_i^{(\ell)} = g(x_i^{(\ell-1)}, y_i^{(\ell)}) := x_i^{(\ell-1)}$ for the first full attention layer as well as its following linear attention layer.

**Lemma B.3.** *The assumptions in* (12) *are satisfied with* $a_1 = 1, a_2 = \cdots = a_L = 2^{3L^2}$.

*Proof of Lemma B.3.* For any $\ell \in [L-1]$, we have

$$Hd(d+1)p(a_1 + a_2 + \cdots + a_{\ell+1}) \leq 2HdpL2^{3L^2} \leq 8^{L^2}(HdpL)^4,$$

while for the right-hand side, we have

$$\sqrt{K}(x_0 \cdots x_{\ell-1}) \cdot (n_1 \cdots n_{L-1}) \geq \sqrt{K} = 8^{L^2}(HdpL)^4 \geq Hd(d+1)p(a_1 + a_2 + \cdots + a_{\ell+1}). \qquad \square$$

**Notation.** For notational convenience, we use $z_{-1}$ and $w$ interchangeably to denote player $-1$'s input. In the following, we elaborate on several key definitions that will be crucial to our proof.

- (Soft transcript $\Pi_{j,i}^{(\ell)}$) For any $i \in [-1 : L-1]$, $j \in [i+1 : L]$, $\ell \in [L]$, recall that $\Pi_{j,i}^{(\ell)}$ denotes the soft transcript sent from player $j$ to player $i$ at the $\ell$-th epoch of communication. Its value is determined by the inputs of players $[i : L]$ (i.e., $z_L, \ldots, z_i$) and is independent of the the inputs of players $[-1 : i-1]$ (i.e., $z_{i-1}, \ldots, z_{-1}$).

  For any fixed inputs $\widetilde{z}_L \in [N_{L-1}]^{N_{L-1}}, \ldots, \widetilde{z}_i \in [N_{i-1}]^{N_{i-1}}$, let $\Pi_{j,i}^{(\ell)}(\widetilde{z}_L, \ldots, \widetilde{z}_i)$ denote the soft transcript when player $t$ receives input $z_t = \widetilde{z}_t$ ($t \in [i : L]$).

- (Linear transcript $\Sigma_{i+1}^{(\ell),m}$) For any $i \in [-1, L-1]$, $l \in [L]$, and $m \in [0, a_\ell - 1]$, recall $\Sigma_{i+1}^{(\ell),m}$ is the linear transcript sent from player $i+1$ to player $i$ in the $(m+1)$-th linear round of the $\ell$-th epoch of communication. Its value is determined by the inputs of players $[i+1 : L]$ (i.e., $z_L, \ldots, z_{i+1}$) and is independent of the the inputs of players $[-1 : i]$ (i.e., $z_i, \ldots, z_{-1}$).

  For any fixed inputs $\widetilde{z}_L \in [N_{L-1}]^{N_{L-1}}, \ldots, \widetilde{z}_i \in [N_{i-1}]^{N_{i-1}}$, let $\Sigma_{i+1}^{(\ell),m}(\widetilde{z}_L, \ldots, \widetilde{z}_i)$ denote the transcript when player $t$ receives input $z_t = \widetilde{z}_t$ ($t \in [i : L]$).

- (The partial composition value) For any $\ell \in [0 : L]$, the value of $i_\ell$ is determined by $w, z_0, \ldots, z_\ell$. We write $i_\ell(\widetilde{w}, \widetilde{z}_0, \ldots, \widetilde{z}_\ell)$ to denote its value when $w = \widetilde{w}, z_0 = \widetilde{z}_0, \ldots, z_\ell = \widetilde{z}_\ell$.

**Indistinguishable decomposition.** Our key concept for the proof is *indistinguishable decomposition* introduced in (Chen et al., 2025). A indistinguishable decomposition is formed by two sets $R_{\geq \ell}$ and $Z_{<\ell}$, where $R_{\geq \ell}$ is a set of input assignments to players $[\ell : L]$ and $Z_{<\ell}$ is a set of input assignments to players $[-1 : \ell-1])$. The key property is that for any fixed input $z_{<\ell} \in Z_{<\ell}$ for the first $\ell$ players, all assignments to $R_{\geq \ell}$ are indistinguishable to players $[-1 : \ell-1]$ on inputs $z_{<\ell}$ after $\ell$ epochs, because they produce identical communication transcripts. Formally, we adapt the definition in (Chen et al., 2025) as follows.

**Definition B.4** (Indistinguishable decomposition). Let $\ell \in [2 : L]$, an indistinguishable decomposition is a pair of sets $R_{\geq \ell} \subseteq A_L \times A_{L-1} \times \cdots \times A_\ell$ and $Z_{<\ell} = Z_{-1} \times \cdots \times Z_{\ell-1}$ with $Z_{-1} = A_{-1}, Z_0 \subseteq A_0, \cdots, Z_{\ell-1} \subseteq A_{\ell-1}$, such that for every $\widetilde{z}_{<\ell} \in Z_{<\ell}$, and for every $\widetilde{\alpha}_{\geq \ell}, \widetilde{\beta}_{\geq \ell} \in R_{\geq \ell}$, it satisfies:

$$\Pi_{j,i}^{(\ell')}(\widetilde{z}_{<\ell}, \widetilde{\alpha}_{\geq \ell}) = \Pi_{j,i}^{(\ell')}(\widetilde{z}_{<\ell}, \widetilde{\beta}_{\geq \ell})$$

for every $j \in [\ell : L]$, $i \in [-1 : \ell-1]$, and $\ell' \in [\ell]$, and

$$\Sigma_{i+1}^{(\ell'),m}(\widetilde{z}_{<\ell}, \widetilde{\alpha}_{\geq \ell}) = \Sigma_{i+1}^{(\ell'),m}(\widetilde{z}_{<\ell}, \widetilde{\beta}_{\geq \ell})$$

for every $i \in [-1 : \ell-1]$, $\ell' \in [\ell]$, and $m \in [0, a_{\ell'} - 1]$.

The utility of an indistinguishable decomposition becomes clear when $\ell = L$. In this case, for every input assignment from $Z_{<L}$ to players $[-1 : L-1]$, player $-1$ (the final output player) observes identical communication transcripts after $L$ epochs (i.e., at the end of the protocol) regardless of which input $\vec{z}_L \in R_{\geq L}$ is assigned to player $L$. Consequently, for every $\widetilde{z}_{<L} \in Z_{<L}$, the output of the protocol $L$-FuncComp$(\widetilde{z}_{<L}, \widetilde{z}_L)$ must be the same for every $\widetilde{z}_L \in R_{\geq L}$. We will carefully define the set $R_{\geq \ell}$ and $Z_{<\ell}$ so that satisfying this requirement leads to a contradiction, thereby establishing the desired lower bound.

For a subset $Z_{<\ell}$, we define the set $\mathcal{I}_{\ell-1}(Z_{<\ell})$ of reachable partial composition values after the $(\ell-1)$-th epoch as

$$\mathcal{I}_{\ell-1}(Z_{<\ell}) := \{i_{\ell-1}(\widetilde{z}_{-1}, \widetilde{z}_0, \ldots, \widetilde{z}_{\ell-1}) | (\widetilde{z}_{-1}, \widetilde{z}_0, \ldots, \widetilde{z}_{\ell-1}) \in Z_{<\ell}\}.$$

In words, $\mathcal{I}_{\ell-1}(Z_{<\ell})$ is the set of all possible values for the intermediate composition when the inputs to players $[-1 : \ell-1]$ are restricted to $Z_{<\ell}$.

The following lemma, as in (Chen et al., 2025), shows that the desired lower bound follows from a good enough indistinguishable configuration for $\ell = L$.

**Lemma B.5.** *An $L$-epoch hybrid communication protocol does not solve $L$-FuncComp if there is an indistinguishable decomposition $R_{\geq L}$ and $Z_{<L}$ with $|R_{\geq L}| \geq |A_L|/\Delta_L$ and $|\mathcal{I}_{L-1}(Z_{<L})| \geq \Theta_{L-1}$.*

The proof of Theorem B.2 is then completed by constructing the required decomposition via an inductive argument, which we will present in the next subsection. This construction is formalized in the following lemma.

**Lemma B.6.** *For every $(L, a_1, \ldots, a_L)$-hybrid communication protocol under assumptions of (12), there is an indistinguishable decomposition $R_{\geq L}$ and $Z_{<L}$ satisfying the requirements of Theorem B.5.*

The remainder of this section is devoted to proving Lemma B.6. We establish a more general inductive claim in Theorem B.7 below. The case $\ell = L$ directly implies Theorem B.6.

**Lemma B.7** (Main Lemma). *For any $\ell \in [2 : L]$, we can construct*

- *a pair of sets $(R_{\geq \ell}, Z_{<\ell})$, where $R_{\geq \ell} \subseteq A_L \times A_{L-1} \times \cdots \times A_\ell$ and $Z_{<\ell} = Z_{-1} \times Z_0 \times \cdots \times Z_{\ell-1}$, with $Z_{-1} = [n_1 \cdots n_{L-1}]$, $Z_0 \subseteq A_0$, $Z_i \subseteq A_i$, and $|Z_i| = x_i$ for $i \in [0 : \ell-1]$.*

- *the soft transcript from players $[\ell : L]$ to $[-1 : \ell-1]$ for the first $\ell$ epochs, when the players $[-1 : \ell-1]$ take input from $Z_{<\ell}$. i.e.,*

$$\Lambda^{(\ell)} := \left(\Lambda_{j,i}^{(\ell,\ell')}\right)_{j\in[\ell:L], i\in[-1:\ell-1], \ell'\in[\ell]}$$

*where*

$$\Lambda_{j,i}^{(\ell,\ell')} := \left(\Lambda_{j,i}^{(\ell,\ell')}(\widetilde{z}_{\ell-1}, \ldots, \widetilde{z}_i)\right)_{\widetilde{z}_{\ell-1}\in Z_{\ell-1}, \ldots, \widetilde{z}_i\in Z_i} \quad \text{and} \quad \Lambda_{j,i}^{(\ell,\ell')}(\widetilde{z}_{\ell-1}, \ldots, \widetilde{z}_i) \in \text{domain}(\Pi_{j,i}^{(\ell')})$$

- *the linear transcript from player $\ell$ to $\ell-1$ for the first $\ell$ epochs, when the players $[-1 : \ell-1]$ take input from $Z_{<\ell}$. i.e.,*

$$\Xi^{(\ell)} := \left(\Xi_\ell^{(\ell'),m}\right)_{\ell'\in[\ell], m\in[0,a_{\ell'}-1]} \quad \text{where} \quad \Xi_\ell^{(\ell'),m} \in \text{domain}(\Sigma_\ell^{(\ell'),m})$$

*such that we have the following properties:*

- *(Consistency of soft transcripts)*

$$\Pi_{j,i}^{(\ell')}(\widetilde{z}_L, \ldots, \widetilde{z}_i) = \Lambda_{j,i}^{(\ell,\ell')}(\widetilde{z}_{\ell-1}, \ldots, \widetilde{z}_i)$$
$$\forall j \in [\ell : L], i \in [-1 : \ell-1], \ell' \in [\ell], \widetilde{z}_{\geq \ell} \in R_{\geq L}, \widetilde{z}_{\ell-1} \in Z_{\ell-1}, \ldots \widetilde{z}_i \in Z_i,$$

- *(Consistency of linear transcripts)*

$$\Sigma_\ell^{(\ell'),m}(\widetilde{z}_L, \ldots, \widetilde{z}_\ell) = \Xi_\ell^{(\ell'),m}, \forall \ell' \in [\ell], m \in [0, a_{\ell'}-1], \widetilde{z}_{\geq \ell} \in R_{\geq L},$$

- *$|R_{\geq \ell}| \geq |A_L| \cdots |A_\ell|/\Delta_\ell$ and $|\mathcal{I}_{\ell-1}(Z_{<\ell})| \geq \Theta_{\ell-1}$.*

## B.2. The Initial step

We first prove Lemma B.7 for the base case $\ell = 2$.

**Step 1: Choosing $Z_0, Z_1$.** We set $Z_0 = [x_0]$. The next step is to select the set $Z_1 \subseteq A_1$. Consider all possible first epoch messages from the player 1 to the player $-1$, denoted by

$$\Psi_{1,-1}^{(1)} = \left( \Psi_{1,-1}^{(1)}(\widetilde{z}_{-1}) \right)_{\widetilde{z}_{-1} \in Z_{-1}} \quad \text{where} \quad \Psi_{1,-1}^{(1)}(\widetilde{z}_{-1}) \in \{0,1\}^{2Hdp}.$$

The total number of of distinct such message patterns $\Psi_{1,-1}^{(1)}$ is $2^{2Hdp \cdot |Z_{-1}|} = 2^{2Hdp(n_1 \cdots n_{L-1})}$. By the pigeonhole principle, there exists a message pattern $\widetilde{\Psi}_{1,-1}^{(1)} \in \{0,1\}^{2Hdp \cdot (n_1 \cdots n_{L-1})}$ such that

$$S := \{\widetilde{z}_1 \in A_1 : \Pi_{1,-1}^{(1)}(\widetilde{z}_1, \widetilde{z}_{-1}) = \Psi_{1,-1}^{(1)}(\widetilde{z}_{-1}) \ \forall \widetilde{z}_{-1} \in Z_{-1}\} \subseteq A_1 \tag{13}$$

satisfies $|S| \geq |A_1| \cdot 2^{-2Hdp \cdot (n_1 \cdots n_{L-1})}$. Note the first epoch message depends only on $\widetilde{z}_1$ and $\widetilde{z}_{-1}$, hence we denote it as $\Pi_{1,-1}^{(1)}(\widetilde{z}_1, \widetilde{z}_{-1})$. The following lemma, as in (Chen et al., 2025), allows us to select a suitable subset $Z_1 \subseteq S$.

**Lemma B.8** ((Chen et al., 2025), Lemma 4.6). *There exists a subset $Z_1 \subseteq S$ with $|Z_1| = x_1$ such that*

$$|\{\widetilde{z}_1(i_0) : \widetilde{z}_1 \in Z_1, i_0 \in Z_0\}| \geq 8^{-L} x_0 x_1 = \Theta_1. \tag{14}$$

We take the subset $Z_1$ given by Lemma B.8. The next step is to fix the transcripts from players $j \in [2 : L-1]$ to players $i = -1, 0, 1$ at the first two epochs.

**Step 2.1: Fixing the transcript to player $-1$.** We begin with the first epoch. Our goal is to fix $\Lambda_{j,-1}^{(2,1)}(\widetilde{z}_1, \widetilde{z}_0, \widetilde{z}_{-1})$ for every $\widetilde{z}_1 \in Z_1, \widetilde{z}_0 \in Z_0, \widetilde{z}_{-1} \in Z_{-1}$. The first epoch message from player $j$ to player $-1$ depends only on $\widetilde{z}_{-1}$ and player $j$'s input, and is independent of $\widetilde{z}_1, \widetilde{z}_0$. Therefore, it suffices to fix a pattern

$$\Phi_{j,-1}^{(1)} = \left( \Phi_{j,-1}^{(1)}(\widetilde{z}_{-1}) \right)_{\widetilde{z}_{-1} \in Z_{-1}} \quad \text{where} \quad \Phi_{j,-1}^{(1)}(\widetilde{z}_{-1}) \in \{0,1\}^{2Hdp}.$$

and then define

$$\Lambda_{j,-1}^{(2,1)}(\widetilde{z}_1, \widetilde{z}_0, \widetilde{z}_{-1}) = \Phi_{j,-1}^{(1)}(\widetilde{z}_{-1}) \quad \forall \widetilde{z}_1 \in Z_1, \widetilde{z}_0 \in Z_0, \widetilde{z}_{-1} \in Z_{-1}$$

The total number of possible such transcript patterns is at most $2^{2Hdp \cdot |Z_{-1}|} = 2^{2Hdp \cdot (n_1 \cdots n_{L-1})}$. By the pigeonhole principle, we can choose a fixed pattern $\{\Lambda_{j,-1}^{(2,1)}\}_{j \in [2:L]}$, such that

$$C_1 := \left\{ \begin{array}{l} (\widetilde{z}_L, \ldots, \widetilde{z}_2) \in A_L \times \cdots \times A_2 : \\ \Pi_{j,-1}^{(1)}(\widetilde{z}_L, \ldots, \widetilde{z}_0, \widetilde{z}_{-1}) = \Lambda_{j,-1}^{(2,1)}(\widetilde{z}_1, \widetilde{z}_0, \widetilde{z}_{-1}) \ \forall \widetilde{z}_1 \in Z_1, \widetilde{z}_0 \in Z_0, \widetilde{z}_{-1} \in Z_{-1}, j \in [2:L] \end{array} \right\}.$$

satisfies $|C_1| \geq |A_L| \cdots |A_2| \cdot 2^{-2Hdp \cdot (n_1 \cdots n_{L-1}) \cdot L}$.

For the second epoch, the goal is to fix $\Lambda_{j,-1}^{(2,2)}(\widetilde{z}_1, \widetilde{z}_0, \widetilde{z}_{-1})$ for every $\widetilde{z}_1 \in Z_1, \widetilde{z}_0 \in Z_0, \widetilde{z}_{-1} \in Z_{-1}$. Crucially, this transcript depends only on the information state $X_{-1}^{(1)}$ and $X_j^{(1)}$, which are themselves independent of the choice of $z_1 \in Z_1$. This independence holds because the only component of $X_{-1}^{(1)}$ and $X_j^{(1)}$ that depends on $z_1$ is the first epoch message from player 1 to $-1$ (recall that $a_1 \leq 1$, (12)), which is fixed to $\Psi_{1,-1}^{(1)}(\widetilde{z}_{-1})$ (13) for all $\widetilde{z}_1 \in Z_1$. Therefore, it suffices to fix

$$\Phi_{j,-1}^{(2)} := \left( \Phi_{j,-1}^{(2)}(\widetilde{z}_0, \widetilde{z}_{-1}) \right)_{\widetilde{z}_0 \in Z_0, \widetilde{z}_{-1} \in Z_{-1}}$$

and then define $\Lambda_{j,-1}^{(2,2)}(\widetilde{z}_1, \widetilde{z}_0, \widetilde{z}_{-1}) = \Phi_{j,-1}^{(2)}(\widetilde{z}_0, \widetilde{z}_{-1}), \forall \widetilde{z}_1 \in Z_1, \widetilde{z}_0 \in Z_0, \widetilde{z}_{-1} \in Z_{-1}$. The total number of transcripts are at most $2^{2Hdp \cdot x_0 \cdot (n_1 \cdots n_{L-1})}$. Therefore, we can choose $\{\Lambda_{j,-1}^{(2,2)}\}_{j \in [2:L-1]}$, such that

$$C_2 := \left\{ \begin{array}{l} (\widetilde{z}_L, \ldots, \widetilde{z}_2) \in C_1 : \\ \Pi_{j,-1}^{(2)}(\widetilde{z}_L, \ldots, \widetilde{z}_0, \widetilde{z}_{-1}) = \Lambda_{j,-1}^{(2,2)}(\widetilde{z}_1, \widetilde{z}_0, \widetilde{z}_{-1}) \ \forall \widetilde{z}_1 \in Z_1, \widetilde{z}_0 \in Z_0, \widetilde{z}_{-1} \in Z_{-1}, j \in [2:L] \end{array} \right\}.$$

satisfies $|C_2| \geq |C_1| \cdot 2^{-2Hdp \cdot x_0(n_1 \cdots n_{L-1}) \cdot L} \geq |A_L \cdots A_2| \cdot 2^{-4HdpL \cdot x_0(n_1 \cdots n_{L-1})}$.

**Step 2.2: Fixing the transcript to player** $0$**.** The total number of $\{\Lambda_{j,0}^{(2,\ell')}\}_{j \in [2:L], \ell' \in [2]}$ is at most $2^{2Hdp \cdot x_0 x_1 \cdot 2L}$. We can fix its value so that

$$C_3 := \left\{ \begin{array}{c} (\widetilde{z}_L, \ldots, \widetilde{z}_2) \in C_2 : \\ \Pi_{j,0}^{(\ell')}(\widetilde{z}_L, \ldots, \widetilde{z}_0) = \Lambda_{j,0}^{(2,\ell')}(\widetilde{z}_1, \widetilde{z}_0) \; \forall \widetilde{z}_1 \in Z_1, \widetilde{z}_0 \in Z_0, j \in [2:L], \ell' \in [2] \end{array} \right\}.$$

satisfies $|C_3| \geq |C_2| \cdot 2^{-2Hdp \cdot x_0 x_1 \cdot 2L} \geq |A_L \cdots A_2| \cdot 2^{-6HdpL \cdot x_0(n_1 \cdots n_{L-1})}$.

**Step 2.3: Fixing the transcript to player** $1$**.** The total number of $\{\Lambda_{j,1}^{(2,\ell')}\}_{j \in [2:L], \ell' \in [2]}$ is at most $2^{2Hdpm \cdot x_1 \cdot 2L}$, and the total number of $\{\Xi_2^{(\ell'),m}\}_{\ell' \in [2], m \in [0, a_{\ell'}-1]}$ is at most $2^{Hd(d+1)p(a_1+a_2)}$, and we can fix the value so that

$$C_4 := \left\{ \begin{array}{l} (\widetilde{z}_L, \ldots, \widetilde{z}_2) \in C_3 : \quad \Sigma_2^{(\ell')}(\widetilde{z}_L, \ldots, \widetilde{z}_1) = \Xi_2^{(\ell')}, \forall \ell' \in [2] \text{ and} \\ \Pi_{j,1}^{(\ell')}(\widetilde{z}_L, \ldots, \widetilde{z}_1) = \Lambda_{j,1}^{(2,\ell')}(\widetilde{z}_1) \; \forall \widetilde{z}_1 \in Z_1, j \in [2:L], \ell' \in [2], \end{array} \right\},$$

satisfies the following bound

$$C_4 \geq C_3 \cdot 2^{-2Hdpm \cdot x_1 \cdot 2L} \cdot 2^{-2Hdp} \geq |A_L| \cdots |A_2| \cdot 2^{-8HdpL \cdot x_0(n_1 \cdots n_{L-1}) - Hd(d+1)p(a_1+a_2)}$$

$$\geq |A_L| \cdots |A_2| \cdot 2^{-4\sqrt{K} x_0(n_1 \cdots n_{L-1})} = |A_L| \cdots |A_2| / \Delta_2 \tag{15}$$

Here, the second step follows from the choice of parameters (see Eq. (6)(8)(12)) and the last step follows from the definition of $\Delta_2$ (see Eq. (10)).

Combining Lemma B.8 and Eq. (15), we conclude the proof for the base case $\ell = 2$.

### B.3. Inductive step

Suppose Theorem B.7 holds up to $\ell \in [2 : L - 1]$. We prove it continues to hold for $\ell + 1$.

The key insight is that $Z_\ell$ is indistinguishable to players $[-1 : \ell - 1]$ after $\ell$ epochs, when these players take input from $Z_{<\ell}$. Hence, the $(\ell + 1)$-th epoch transcripts to players $[-1 : \ell - 1]$ are independent of the choice of $z_\ell \in Z_\ell$.

**Step 1: Choosing the set** $Z_\ell$**.** Recall that the size of $R_{\geq \ell}$ satisfies $|R_{\geq \ell}| \geq |A_L| \times \cdots \times |A_\ell| / \Delta_\ell$. We would like to select a rectangular subset from $R_{\geq \ell}$. As in (Chen et al., 2025), we have the following lemma.

**Lemma B.9** ((Chen et al., 2025), Lemma 4.7)**.** *There exists a subset $S^{(\ell)} \subseteq R_{\geq \ell}$ such that*

- $S^{(\ell)} = S_1^{(\ell)} \times S_2^{(\ell)}$, *where* $S_1^{(\ell)} \subseteq A_L \times \cdots \times A_{\ell+1}$, $S_2^{(\ell)} \subseteq A_\ell$, *such that*

$$|S_1^{(\ell)}| \geq |A_L| \cdots |A_{\ell+1}| / \Delta_\ell^{2x_\ell} \quad and \quad |S_2^{(\ell)}| = x_\ell.$$

- $|\{i_\ell : i_\ell = \widetilde{z}_\ell(\widetilde{w}_{\ell-1}, \widetilde{i}_{\ell-1}) \text{ for some } \widetilde{w}_{\ell-1} \in [n_{\ell-1}], \widetilde{i}_{\ell-1} \in \mathcal{I}_{\ell-1}, \widetilde{z}_\ell \in S_2^{(\ell)}\}| \geq \Theta_\ell.$

With Lemma B.9 in hand, we take $Z_\ell = S_2^{(\ell)}$ and $S_{\geq \ell+1} = S_1^{(\ell)}$.

Next, we are going to fix the transcript $\Lambda^{(\ell+1)}$. Recall that we need to fix all transcripts from players $[\ell + 1 : L]$ to players $[-1 : \ell]$ in the first $\ell + 1$ epochs, when players $[-1 : \ell]$ receive input from $Z_{\leq \ell} = Z_{-1} \times Z_0 \times \cdots \times Z_\ell$. We proceed in a few steps.

**Step 2.1: Fixing the transcript to players** $[-1 : \ell - 1]$ **in the first** $\ell$ **epochs.** We simply use $\Lambda^{(\ell)}$, that is, for any $\widetilde{z}_\ell \in Z_\ell, \ldots, \widetilde{z}_i \in Z_i$,

$$\Lambda_{j,i}^{(\ell+1,\ell')}(\widetilde{z}_\ell, \ldots, \widetilde{z}_i) = \Lambda_{j,i}^{(\ell,\ell')}(\widetilde{z}_{\ell-1}, \ldots, \widetilde{z}_i). \quad \forall j \in [\ell + 1 : L], i \in [-1 : \ell - 1], \ell' \in [\ell] \tag{16}$$

As in (Chen et al., 2025), $S_{\geq \ell+1} \subseteq A_L \times \cdots \times A_{\ell+1}$ is consistent with $\Lambda^{(\ell+1)}$ up to this point.

**Lemma B.10.** *The set $S_{\geq \ell+1}$ is consistent with $\{\Lambda_{j,i}^{(\ell,\ell')}\}_{j\in[\ell+1:L], i\in[-1:\ell-1], \ell'\in[\ell]}$. Formally, for any $\widetilde{z}_{\geq \ell+1} \in S_{\geq \ell+1}$ and $\widetilde{z}_{<\ell+1} \in Z_{<\ell+1}$, one has*

$$\Pi_{j,i}^{(\ell')}(\widetilde{z}_L, \ldots, \widetilde{z}_i) = \Lambda_{j,i}^{(\ell+1,\ell')}(\widetilde{z}_\ell, \ldots, \widetilde{z}_i).$$

*for any $j \in [\ell+1:L], i \in [-1:\ell-1], \ell' \in [\ell]$.*

**Step 2.2: Fixing the transcript to players $[-1:\ell-1]$ at the $(\ell+1)$-th epoch.**  Our key insight is that $Z_\ell$ is indistinguishable to players $[-1:\ell-1]$ when they take input from $Z_{\leq \ell-1}$. Hence, their transcripts are independent of the choice of $z_\ell \in Z_\ell$. We consider

$$\Phi^{(\ell+1)} = \left(\Phi_{j,i}^{(\ell+1)}\right)_{j\in[\ell+1:L], i\in[-1:\ell-1]}$$

where

$$\Phi_{j,i}^{(\ell+1)} = \left(\Phi_{j,i}^{(\ell+1)}(\widetilde{z}_{\ell-1}, \ldots, \widetilde{z}_i)\right)_{\widetilde{z}_{\ell-1}\in Z_\ell \ldots, \widetilde{z}_i\in Z_i} \quad \text{and} \quad \Phi_{j,i}^{(\ell+1)}(\widetilde{z}_{\ell-1}, \ldots, \widetilde{z}_i) \in \text{domain}(\Pi_{j,i}^{(\ell+1)})$$

Comparing $\Lambda_{j,i}^{(\ell+1,\ell+1)}$ with $\Phi_{j,i}^{(\ell+1)}$, we note that $\Phi_{j,i}^{(\ell+1)}$ is independent of $\widetilde{z}_\ell \in Z_\ell$. For any $\Phi^{(\ell+1)}$, define

$$S(\Phi^{(\ell+1)}) := \left\{ \begin{array}{c} (\widetilde{z}_L, \ldots, \widetilde{z}_{\ell+1}) \in S_{\geq \ell+1}: \\ \Pi_{j,i}^{(\ell+1)}(\widetilde{z}_L, \ldots, \widetilde{z}_i) = \Phi_{j,i}^{(\ell+1)}(\widetilde{z}_{\ell-1}, \ldots, \widetilde{z}_i) \\ \forall \widetilde{z}_\ell \in Z_\ell, \ldots, \widetilde{z}_i \in Z_i, j \in [\ell+1:L], i \in [-1:\ell-1] \end{array} \right\} \tag{17}$$

In words, $S(\Phi^{(\ell+1)})$ includes all $(\widetilde{z}_L, \ldots, \widetilde{z}_{\ell+1}) \in S_{\geq \ell+1}$ that are consistent with the transcript $\Phi^{(\ell+1)}$. The proof of the following lemma differs from (Chen et al., 2025) because in our hybrid communication model, we must account for linear transcripts.

**Lemma B.11.** *We have*

$$\bigcup_{\Phi^{(\ell+1)}} S(\Phi^{(\ell+1)}) = S_{\geq \ell+1}.$$

*Proof.* It suffices to prove that, for any $(\widetilde{z}_L, \ldots, \widetilde{z}_{\ell+1}) \in S_{\geq \ell+1}, \widetilde{z}_{\ell-1} \in Z_{\ell-1}, \ldots, \widetilde{z}_i \in Z_i, j \in [\ell+1:L], i \in [-1:\ell-1]$, the transcript $\Pi_{j,i}^{(\ell+1)}(\widetilde{z}_L, \ldots, \widetilde{z}_{\ell+1}, z_\ell, \widetilde{z}_{\ell-1}, \ldots \widetilde{z}_i)$ is the same for every $z_\ell \in Z_\ell$.

To prove this, note that the transcript $\Pi_{j,i}^{(\ell+1)}$ is determined by the information states $X_j^{(\ell)}$ and $X_i^{(\ell)}$. It is clear that $X_j^{(\ell)}$ does not change with the choice of $z_\ell \in Z_\ell$ since $j > \ell$. It remains to prove that $X_i^{(\ell)}$ also does not change with $z_\ell \in Z_\ell$. We prove that all information states $\{X_r^{(\ell'),m}\}_{r\in[i:\ell-1], \ell'\in[\ell]}$ do not change with $z_\ell$.

Recall we have fixed the values of $(\widetilde{z}_L, \ldots, \widetilde{z}_{\ell+1}) \in S_{\geq \ell+1}$ and $\widetilde{z}_{\ell-1} \in Z_{\ell-1}, \ldots, \widetilde{z}_i \in Z_i$. We prove this by downward induction on $r$, from $r = \ell-1$ to $r = i$. When $r = \ell-1$, the information state $X_{\ell-1}^{(\ell')}$ is determined by $\widetilde{z}_{\ell-1}$, $\Pi_{t,\ell-1}^{(\ell'')}(\widetilde{z}_L, \ldots, \widetilde{z}_{\ell+1}, z_\ell, \widetilde{z}_{\ell-1})$, and $\Sigma_\ell^{(\ell''),m}(\widetilde{z}_L, \ldots, \widetilde{z}_{\ell+1}, z_\ell, \widetilde{z}_{\ell-1})$ $(t \in [\ell:L], \ell'' \in [\ell'], m \in [0, a_{\ell''}-1])$.

Since $(\widetilde{z}_L, \ldots, \widetilde{z}_{\ell+1}, z_\ell) \in R_{\geq \ell}$ for every $z_\ell \in Z_\ell$, we have that $\Pi_{t,\ell-1}^{(\ell'')}(\widetilde{z}_L, \ldots, \widetilde{z}_{\ell+1}, z_\ell, \widetilde{z}_{\ell-1}) = \Lambda_{t,\ell-1}^{(\ell,\ell'')}(\widetilde{z}_{\ell-1})$ and $\Sigma_\ell^{(\ell''),m}(\widetilde{z}_L, \ldots, \widetilde{z}_{\ell+1}, z_\ell, \widetilde{z}_{\ell-1}) = \Xi_\ell^{(\ell''),m}(\widetilde{z}_{\ell-1})$, which are the same for every $z_\ell \in Z_\ell$. This finishes the proof of the base case. Now suppose the assertion holds for $r+1$. Then, for $r$, $X_r^{(\ell')}$ is determined by $\widetilde{z}_r, \Pi_{t,r}^{(\ell'')}(\widetilde{z}_L, \ldots, \widetilde{z}_{\ell+1}, z_\ell, \widetilde{z}_{\ell-1}, \ldots, \widetilde{z}_r)$, and $\Sigma_{r+1}^{(\ell''),m}(\widetilde{z}_L, \ldots, \widetilde{z}_{\ell+1}, z_\ell, \widetilde{z}_{\ell-1}, \ldots, \widetilde{z}_r)$ $(t \in [r+1:L], \ell'' \in [\ell'], m \in [0, a_{\ell''}-1])$.

For $t \in [\ell:L]$, since $(\widetilde{z}_L, \ldots, \widetilde{z}_{\ell+1}, z_\ell) \in R_{\geq \ell}$, we have $\Pi_{t,r}^{(\ell'')}(\widetilde{z}_L, \ldots, \widetilde{z}_{\ell+1}, z_\ell, \widetilde{z}_{\ell-1}, \ldots, \widetilde{z}_r) = \Lambda_{t,r}^{(\ell,\ell'')}(\widetilde{z}_{\ell-1}, \ldots, \widetilde{z}_r)$, which is the same for every $z_\ell \in Z_\ell$. For $t \in [r+1:\ell-1]$, we have proved that $X_t^{(\ell'')}$ are the same for every $z_\ell \in Z_\ell$, so does $\Pi_{t,r}^{(\ell'')}(\widetilde{z}_L, \ldots, \widetilde{z}_{\ell+1}, z_\ell, \widetilde{z}_{\ell-1}, \ldots, \widetilde{z}_r)$. Moreover, $\Sigma_{r+1}^{(\ell''),m}(\widetilde{z}_L, \ldots, \widetilde{z}_{\ell+1}, z_\ell, \widetilde{z}_{\ell-1}, \ldots, \widetilde{z}_r)$ depends only on $X_{r+1}^{(\ell''),m}$, which is the same for every $z_\ell \in Z_\ell$. This completes the induction and finishes the proof. $\square$

Now as in (Chen et al., 2025), we obtain

**Lemma B.12.** *There exists $\widetilde{\Phi}^{(\ell+1)}$ such that*

$$|S(\widetilde{\Phi}^{(\ell+1)})| \geq |A_L| \cdots |A_{\ell+1}| \cdot 2^{-2\sqrt{K} \cdot (x_0 \cdots x_{\ell-1}) \cdot (n_1 \cdots n_{L-1})}$$

*The set $T_{\geq \ell+1} = S(\widetilde{\Phi}^{(\ell+1)}) \subseteq S_{\geq \ell+1}$ is consistent with the transcripts $(\Lambda_{j,i}^{(\ell+1,\ell')})_{i \in [-1:\ell-1], j \in [\ell+1:L], \ell' \in [\ell+1]}$. Formally, for any $(\widetilde{z}_L, \ldots, \widetilde{z}_{\ell+1}) \in T_{\geq \ell+1}$ and $\widetilde{z}_{<\ell+1} \in Z_{<\ell+1}$, one has*

$$\Pi_{j,i}^{(\ell')}(\widetilde{z}_L, \ldots, \widetilde{z}_i) = \Lambda_{j,i}^{(\ell+1,\ell')}(\widetilde{z}_\ell, \ldots, \widetilde{z}_i). \tag{18}$$

*for any $j \in [\ell+1:L], i \in [-1:\ell-1], \ell' \in [\ell+1]$. Moreover, we have*

$$|T_{\geq \ell+1}| \geq |A_L \times \cdots \times A_{\ell+1}| \cdot 2^{-2\sqrt{K} \cdot (x_0 \cdots x_{\ell-1}) \cdot (n_1 \cdots n_{L-1})}. \tag{19}$$

**Step 2.3: Fixing the transcript to player $\ell$.** This follows a greedy selection strategy. Let

$$\Psi = \left( \Psi_{j,\ell}^{(\ell')}(\widetilde{z}_\ell) \right)_{j \in [\ell+1:L], \ell' \in [\ell+1], \widetilde{z}_\ell \in Z_\ell} \quad \text{where} \quad \Psi_{j,\ell}^{(\ell')}(\widetilde{z}_\ell) \in \mathsf{domain}(\Pi_{j,\ell}^{(\ell')})$$

Define

$$T(\Psi) := \left\{ \begin{array}{c} (\widetilde{z}_L, \ldots, \widetilde{z}_{\ell+1}) \in T_{\geq \ell+1} : \\ \Pi_{j,\ell}^{(\ell')}(\widetilde{z}_L, \ldots, \widetilde{z}_\ell) = \Psi_{j,\ell}^{(\ell')}(\widetilde{z}_\ell) \quad \forall \widetilde{z}_\ell \in Z_\ell, \ell' \in [\ell+1], j \in [\ell+1:L] \end{array} \right\} \tag{20}$$

As in (Chen et al., 2025), we can upper bound the number of different $\Psi$ and use the pigeonhole principle to obtain the following lemma.

**Lemma B.13.** *The total number of $\Psi$ is at most $2^{\sqrt{K} \cdot (x_0 \cdots x_{\ell-1}) \cdot (n_1 \cdots n_{L-1})}$. Hence, there exists $\widetilde{\Psi}$ such that $|T(\widetilde{\Psi})| \geq |A_L| \cdots |A_{\ell+1}| 2^{-3\sqrt{K} \cdot (x_0 \cdots x_{\ell-1}) \cdot (n_1 \cdots n_{L-1})}$.*

Given Lemma B.13, we fix the transcripts from players $j \in [\ell+1:L]$ to players $\ell$ during the first $\ell+1$ epochs using $\widetilde{\Psi}$. In particular, we take

$$\Lambda_{j,\ell}^{(\ell+1,\ell')}(\widetilde{z}_\ell) = \widetilde{\Psi}_{j,\ell}^{(\ell')}(\widetilde{z}_\ell), \quad \forall j \in [\ell+1:L], \ell' \in [\ell+1], \widetilde{z}_i \in Z_i. \tag{21}$$

Next, we fix $\left( \Xi_{\ell+1}^{\ell'} \right)_{\ell' \in [\ell+1]}$. The total number of choices is $2^{2Hdp(a_1+a_2+\cdots+a_{\ell+1})}$, so there exists a choice for which the size of its consistency set is at least

$$|A_L| \cdots |A_{\ell+1}| \cdot 2^{-3\sqrt{K} \cdot (x_0 \cdots x_{\ell-1}) \cdot (n_1 \cdots n_{L-1}) - Hd(d+1)p(a_1+a_2+\cdots+a_{\ell+1})} \geq |A_L| \cdots |A_{\ell+1}|/\Delta_{\ell+1}$$

by (12). We can then take $R_{\geq \ell+1}$ to be this consistency set, and this completes the induction step.

## C. Upper bound for full attention

This section provides constructive proofs for the upper bounds stated in Theorems A.3, A.6, A.5, and A.8. Specifically, we design Transformer decoders equipped with full attention that successfully solve the respective tasks. Central to our design is a retrieval head mechanism, adapted from (Chen et al., 2025). We note that an alternative, more implicit construction leveraging nearly orthogonal vectors (Bhattamishra et al., 2024) could similarly be employed.

---

**Retrieval task**

**Input.** For the first $n$ input tokens $x_1, \cdots, x_n$, each token $x_i$ is composed of vectors $a_i \in \{0,1\}^D$ and $b_i \in \{0,1\}^D$, and the last token $x_{n+1}$ contains a query vector $a$.
**Task.** Find the position $i \in [n]$ such that $a_i = a$, and return the corresponding value $b_i$.
**Output.** If a unique $i$ satisfies $a_i = a$, output $b_i$. Otherwise, the output can be arbitrary.

---

We now detail the implementation of this retrieval operation using a single attention head.

*Implementation of the retrieval head.* We set the value projection $V$ to be $b_i$, and the key projection $K$ for position $i \in [n]$ to be $\log^2(n) \cdot (a_i, \vec{1} - a_i)$ for position $i \in [n]$, where $\vec{1} \in \{0,1\}^D$ denotes the all-ones vector of length $D$, and $\vec{1} - a_i$ denotes element-wise subtraction; the query projection $Q$ at position $n+1$ is taken to be $\log^2(n) \cdot (a, \vec{1} - a)$. The attention score (before softmax) satisfies

$$\langle Qx_{n+1}^{(\ell)}, Kx_i^{(\ell)} \rangle = \begin{cases} \log^2(n)D & a_i = a \\ \leq \log^2(n)D - \log^2(n) & a_i \neq a. \end{cases}$$

Hence, if there is exactly one position $i \in [n]$ that satisfies $a_i = a$, then the attention probabilities satisfy

$$\alpha_{n+1,i} \geq \frac{\exp(\log^2(n))}{\exp(\log^2(n)) + n - 1} \geq 1 - \frac{n}{n^{\log(n)}},$$

which is indistinguishable from 1 under the assumption of precision $p = \Theta(\log(n))$, and

$$\alpha_{n+1,j} \leq \frac{1}{n^{\log(n)}},$$

for all $j \neq i$, which is indistinguishable from 0 under the assumption of precision $p = \Theta(\log(n))$. We conclude that, under $p = \Theta(\log n)$-bit precision, the attention head will attend exclusively to position $i$ and retrieve the value $b_i$. $\qquad \square$

Below, we explain how this mechanism enables the solution for Eva and PerCom.

**Construction for Eva.** The function evaluation task can be formulated as a retrieval task. Here, the last token of the input represents the query $a = x \in [n]$. For each preceding position $i \in [n]$.(whose token encodes the value $f(i)$), we set $a_i = i$ and $b_i = f(i)$. The objective is to identify the unique position $i$ satisfying $i = x$ and output $f(i)$. Applying the retrieval head from the previous subsection directly yields $f(x)$.

**Construction for PerCom.** PerCom can be viewed as $n$ retrieval tasks. The final $n$ tokens of the input represent the elements $\tau(1), \cdots, \tau(n)$. Each of these tokens provides a query $a$ for the retrieval operation at its respective position. For every preceding token $j \in [n]$, the $j$-th token corresponds to the $\sigma(j)$, and it sets $a_j = j$ and $b_j = \sigma(j)$. To calculate $\sigma(\tau(i))$, the model performs the retrieval task to find the unique position $j \in [n]$ such that $j = \tau(i)$ and then outputs the value $\sigma(j)$. Therefore, the retrieval head implementation accomplishes PerCom.

## D. Some Missing Proof

*Proof of Lemma 2.2.* Suppose that we have a linear attention head of dimension $d$ and precision $p$, with query, key, and value matrices $Q, K$, and $V$. Let $S_i = S_{i-1} + Vx_i \otimes \varphi(Kx_i)$, $Z_i = Z_{i-1} + \varphi(Kx_i)$ with $S_0 = 0$ and $Z_0 = 0$, we have

$$y_i = \frac{\varphi(Qx_i)^\top S_i}{\varphi(Qx_i)^\top Z_i}.$$

Let $h_0 = (S_0, Z_0) \in \mathbb{R}^{2d}$, $g(x, h = (S, Z)) = (S + Vx \otimes \varphi(Kx), Z + \varphi(Kx))$ and

$$f(x, h = (S, Z)) = \frac{\varphi(Qx)^\top S}{\varphi(Qx)^\top Z}$$

where both $g$ and $f$ are independent of the time step $i$, one easily verifies that the corresponding RNN computes the same output as the given linear attention layer. $\qquad \square$

## E. Limitations and Discussion

### E.1. Limitations

Our theoretical analysis establishes lower bounds for two specific classes of efficient attention: **Linear-Full hybrid attention** and **Block-sparse attention**. Many practical efficient attention designs, however, fall *outside* these classes and are therefore **not** subject to our impossibility results. For instance, DeepSeek's DSA (Deepseek sparse attention (DeepSeek-AI, 2025)) uses a learnable indexer to select individual tokens per query without block compression. This fine-grained, content-aware

routing avoids the block-compression bottleneck that underpins our sparse attention lower bound (Theorem 2.8). Our results therefore do **not** apply to DSA or similar learnable sparse patterns.

Moreover, some practical hybrid models, such as Kimi Linear (Team et al., 2025), interleave linear and full attention not at the layer level but **within the same layer across different heads**. In these designs, some attention heads use full softmax attention while others use linear (or linear-like) recurrences, and the outputs are concatenated. This head-wise hybridization is qualitatively different from the layer-wise block structure we analyzed. Our lower bounds do **not** directly extend to head-wise hybrids. Analyzing whether and under what conditions head-wise mixing can overcome the limitations we proved is an interesting open question.

Other architectural innovations that may circumvent our results include:

- **Adaptive per-token routing** – where the model decides dynamically, based on the input, whether to use a full-attention, linear-attention, or other primitive for each token (or each step). Because our communication model assumes a fixed pattern of full-attention and linear-attention layers, adaptive routing can potentially allocate full-attention capacity precisely where it is most needed, thereby breaking the uniform layer-counting argument.

- **Global memory tokens** – extra learnable tokens that persist across the sequence and can be attended to by any position. Such tokens can act as a shared workspace, reducing the need for deep sequential composition within the recurrent state. Our lower bounds do not account for external memory mechanisms, and thus models leveraging them may achieve high performance even with few full-attention layers.

Our lower bounds identify provable weakness of (i) a pure layer-wise linear-full hybrid attention, or (ii) a block-compression sparse pattern. Architectures that avoid these conditions—through learnable per-token selection, head-wise hybridization, adaptive routing, memory tokens, or other mechanisms—remain outside the scope of our impossibility results. We view the development of such architectures as an exciting and complementary line of research. Our theoretical framework provides a lens for understanding which design choices are essential for deep compositional reasoning, and we hope it will guide the creation of even more efficient yet expressive attention mechanisms in the future.

### E.2. Choice of Tasks

The two tasks studied in this paper—$L$-sequential function composition and 2-Sum—are deliberately synthetic. We now clarify why these specific tasks are well-suited for our theoretical analysis and what limitations they carry.

**Why these tasks?** The $L$-sequential function composition task is designed to isolate the core challenge of *dependent, non-parallelizable steps* within a single forward pass. Each step's output defines the input for the next, mirroring the structure of multi-hop retrieval, reasoning chains, or algorithmic execution—tasks where a model cannot rely on parallel shortcuts. This structure allows us to apply a communication complexity argument, where information must propagate sequentially through the layers. The clean separation between full attention (which can aggregate globally) and linear recurrent layers (which propagate information only through a bounded hidden state) becomes starkly visible on this task.

The 2-Sum task, by contrast, captures the need for *pairwise comparisons* across the entire sequence. It is simple enough to analyze but requires the model to detect a global pattern (two tokens summing to zero). This makes it a natural testbed for sparse attention: any mechanism that compresses blocks loses the ability to distinguish certain pairs unless the compression is sufficiently rich. The separation we prove shows that block-compression sparse attention must pay a price in capacity that full attention avoids.

**Limitations.** The precise hierarchical composition in our $L$-sequential task (with carefully constrained retrieval scopes) is unlikely to appear verbatim in natural language or real-world data. Real-world multi-hop reasoning is often less structured and may allow shortcuts or parallel processing that our synthetic task explicitly forbids. Moreover, both tasks are far simpler than the complex, open-ended problems that modern LLMs tackle. Our results should be interpreted as existence proofs of *separation* rather than as direct predictors of performance on benchmarks.

**Bridging the gap to practice.** Despite these limitations, we believe the tasks capture essential difficulty that does arise in practical scenarios. For example, multi-hop question answering (e.g., "Who is the mother of the person who wrote ...?") requires sequential composition of retrieval steps. Logical reasoning (e.g., "If A then B, and B then C, ...") likewise involves

dependent steps. The 2-Sum task is a proxy for tasks requiring global pairwise comparisons, such as duplicate detection or certain forms of entity matching.

Our goal is not to claim that hybrid models fail on all real tasks, but to establish a formal boundary: *there exists* a family of natural (if synthetic) tasks where the separation is provable. We hope that future empirical work will test whether this theoretical separation manifests in practical settings, and we have suggested concrete testable hypotheses (e.g., comparing hybrid models with varying numbers of full-attention layers on multi-hop retrieval benchmarks).

In summary, the tasks are chosen for analytical tractability, not for direct emulation of real-world data. They serve as a rigorous proving ground for understanding fundamental limits, and we believe the insights they provide are valuable even outside the synthetic setting.

