# OpenReview forum: "A Provable Expressiveness Hierarchy in Hybrid Linear-Full Attention"
_ICML.cc/2026/Conference — ICML 2026 regular_

### Official Review · Reviewer_73Jm · 2026-02-20

**Soundness:** 3
**Presentation:** 3
**Significance:** 3
**Originality:** 3
**Overall Recommendation:** 3
**Confidence:** 3

**Summary:**

This work studies the theoretical expressive power of efficient attention mechanisms relative to standard softmax full attention in decoder-only Transformers. It formalizes a class of hybrid linear–full attention architectures (full-attention layers interleaved with blocks of linear-attention layers.

The main result is a provable separation: an $(L{+}1)$-layer full-attention Transformer can solve the task, while a hybrid architecture with only $L{-}1$ full-attention layers cannot, even if it includes an exponentially large number of linear-attention layers, under stated size/precision constraints. The paper also proves a limitation for a class of single-layer block sparse attention (block compression + block selection) on the 2-Sum task, showing that substantially larger capacity is required compared to full attention. The technical approach is based on communication-complexity-style models and an adaptation of the “indistinguishable decomposition” technique used in prior work

**Compliance With Llm Reviewing Policy:**

Affirmed.

**Final Justification:**

most of my concerns are resolved. For more details, please refer to the rebuttal acknowledge section.

**Key Questions For Authors:**

please refer to the weakness section

**Limitations:**

yes

**Strengths And Weaknesses:**

Strength

- The paper tackles a well-motivated open theoretical question: how the expressiveness of linear / hybrid / sparse attention compares to softmax full attention.
- The hybrid-attention lower bound is built on a clear formalization pipeline: define the architectures (full, linear-as-recurrence, hybrid), define the target task (L-sequential function composition), introduce a tailored hybrid communication model, then prove a separation using an indistinguishability argument.
- A provable separation between hybrid linear–full attention and full attention, even on a synthetic task, is a meaningful contribution to the theory of Transformers and efficient attention—especially given the popularity of hybrid designs in practice.



Weakness

1. The primary separation is shown on highly structured synthetic tasks (sequential function composition; 2-Sum). The manuscript itself notes that the real-world implications “demand verification in practice,” but the paper does not include any empirical validation or even small-scale experiments demonstrating that the identified limitation correlates with behavior of modern hybrid-attention LLMs. This limits how strongly one can translate the theorem into architectural guidance.
2. The hybrid lower bound involves parameter regimes and layer counts that are far from typical practice (e.g., exponentially many linear layers relative to the number of full layers). While this is theoretically informative, the paper could do more to interpret what the result suggests for realistic scales (e.g., constant $L$, moderate context lengths, practical $H,d,p$).
3. The hybrid architecture definition is specific (blocks of linear layers after each full-attention layer). Many practical hybrids include additional mechanisms (gating, memory tokens, cross-layer KV reuse, specialized residual pathways, mixture-of-experts routing, etc.). It is not fully clear which of these would be captured by the current “linear attention as recurrence” abstraction and which could plausibly circumvent the lower bound.
4. The sparse-attention theorem is restricted to single-layer attention with a particular block compression/selection formulation. This is a meaningful first step, but it does not directly cover multi-layer sparse Transformers or other sparse patterns (e.g., sliding windows + global tokens), which are common in long-context models.

---

> ### Author Rebuttal · Authors · 2026-03-31
>
> We truly appreciate the time and care you’ve invested in reviewing our work and your constructive feedback.
>
> 1. Synthetic tasks and lack of empirical validation.
>
> We completely agree that the tasks are synthetic - this is by design, allowing us to perform a clean, rigorous information-theoretic analysis. The sequential function composition task captures the core structure of multi-step reasoning (e.g., multi-hop retrieval, logical chains), and such synthetic tasks are widely adopted in theoretical analysis (e.g., Chen et al., 2025; Sanford et al., 2023). Our contribution is foundational, and we are genuinely excited to see future empirical work build on it.
>
> 2. Exponential number of linear layers and parameter regimes.
>
> The exponential dependence on L serves a qualitative purpose: it shows that the separation is fundamental - even an astronomically large number of linear layers cannot compensate for one fewer full-attention layer. For realistic $L$ (e.g., 10), the number $2^{3L^2}$ is astronomically large, but the result establishes that the limitation is not merely polynomial.
>
> 3. Restrictive hybrid architecture definition.
>
> Our hybrid model is a clean abstraction. We fully acknowledge that many practical hybrids (e.g., head-wise mixing, memory tokens, adaptive routing) may circumvent our lower bound. We acknowledge this as a limitation of our current framework, and we see it as an exciting avenue for future work to extend the analysis to more sophisticated architectures.
>
> 4. Sparse attention scope.
>
> The sparse attention theorem focuses on a specific block-compression + selection pattern - a common building block in sparse attention. It does not cover other patterns (e.g., global tokens, learnable routing), and we are careful to note this. Extending the analysis to broader sparse families is another exciting direction we hope to explore.
>
> We are genuinely grateful for your constructive engagement.

---

> > ### Author Rebuttal · Reviewer_73Jm · 2026-04-05
> >
> > The rebuttal is helpful and I appreciate the authors’ clear clarifications. In particular, the authors explicitly acknowledge that the synthetic tasks, the specific hybrid abstraction, and the restricted sparse-attention setting are deliberate choices made to enable a clean theoretical analysis.
> >
> > However, my main concerns remain only partially addressed.
> > The central issue is not whether these limitations are acknowledged, but whether the current results are sufficient to support strong conclusions about practically relevant efficient-attention architectures. The main separations are shown on highly structured synthetic tasks, under architecture definitions and parameter regimes that are still somewhat far from common practice. As a result, it remains unclear how much architectural guidance these theorems currently provide beyond the formal settings studied in the paper.
> >
> > I agree that this is a meaningful theoretical step, and I appreciate the authors’ careful positioning of the work as foundational. At the same time, I still feel that the gap between the theorem statements and their practical implications is substantial enough that it would require a more significant revision—rather than a short rebuttal—to fully address.
> >
> > Based on the rebuttal, I am inclined to keep my overall assessment near my original score.

---

> > > ### Author Response · Authors · 2026-04-05
> > >
> > > Thank you for your continued engagement and for acknowledging the clarity of our rebuttal. We fully understand your concern that the gap between our theoretical setting and practical architectures remains substantial. We would like to offer a few final perspectives.
> > >
> > > First, the role of such impossibility results.
> > >
> > > We view our work as establishing a boundary condition: no matter how many linear layers one adds (even exponentially many), a single missing full-attention layer creates a fundamental limitation on tasks that require deep compositional reasoning. The synthetic task is carefully constructed to isolate this property. While the parameter regime is far from practice, the qualitative message - that linear attention cannot substitute for full attention in certain compositional scenarios - is, we believe, both theoretically sound and practically suggestive.
> > >
> > > Second, bridging the gap.
> > >
> > > In response to Reviewer A2RP, we have already committed to adding a dedicated discussion (in the conclusion or an appendix, clearly referenced in the main text) that surveys existing efficient designs (e.g., DeepSeek's DSA, head‑wise hybrids, adaptive routing) that fall outside our lower bounds. This will explicitly contextualize our results and help readers understand which architectural choices can circumvent the identified bottlenecks. We believe this addition will substantially improve the paper’s practical relevance.
> > >
> > > Third, a suggestion for empirical translation.
> > >
> > > Our result implies a concrete, testable hypothesis: on multi‑step reasoning tasks (e.g., multi-hop retrieval or reasoning chains), models with fewer full-attention layers - even when augmented with many linear layers - should underperform those with one additional full-attention layer, all else being equal. We hope this may inspire empirical work that directly validates or refines our theoretical predictions.
> > >
> > > We are grateful for your thoughtful critique, which has helped us sharpen the paper’s positioning. We hope that our commitment to adding the practical discussion, together with the above clarifications, might encourage you to reconsider your assessment. Thank you again for your valuable input and we will fully incorporate it in the final revision.

---

### Official Review · Reviewer_KkDZ · 2026-03-09

**Soundness:** 2
**Presentation:** 2
**Significance:** 2
**Originality:** 2
**Overall Recommendation:** 4
**Confidence:** 3

**Summary:**

This paper studies the expressive power of hybrid attention architectures that interleave full (softmax) attention layers with linear attention layers. The main result (Theorem 1.1) shows that for the L-sequential function composition task, an (L+1)-layer full attention network suffices, but any hybrid network with L-1 full attention layers cannot solve it even when augmented with an exponentially large number $(2^{3L^2})$ of linear attention layers. The paper also proves a separation between sparse attention and full attention on the 2-Sum task (Theorem 1.2). The proof methodology extends the communication complexity framework and indistinguishable decomposition technique from Chen et al. (FOCS 2025) to handle the additional information flow introduced by linear attention layers in the hybrid setting. The appendix further includes lower bounds for pure linear attention and log-linear attention on function evaluation and permutation composition tasks.

**Compliance With Llm Reviewing Policy:**

Affirmed.

**Final Justification:**

Most of my concerns are solved.

**Key Questions For Authors:**

- Can you clarify the exact sense in which your linear attention formulation covers Mamba? Mamba uses selective state spaces with data-dependent gating of the A, B, C matrices, which gives it more expressive hidden state dynamics than the simple additive outer-product recurrence in your Eq. for $S_i$. When you invoke Lemma 2.2, what hidden dimension do you get for Mamba specifically?
- Your lower bound shows that $2^{3L^2}$ linear attention layers don't help. But practical systems use far fewer linear layers (e.g., 3:1 ratio). Is there any hope of strengthening the result to show that even a polynomial number of linear layers doesn't help, or is the exponential bound tight?

**Limitations:**

The authors acknowledge the theoretical nature of the task. However, they do not adequately discuss (a) the gap between their architectural model and practical hybrid designs, (b) the parameter regimes where the result is meaningful, or (c) any potential negative societal impacts (though for a pure theory paper this is less concerning). The paper would benefit from a more candid discussion of the gap between the theoretical separation and practical observations that hybrid models work quite well.

**Strengths And Weaknesses:**

### Strengths
- S1. Timely and well-motivated question. The hybrid linear-full attention architecture has become a hot topic in practice  -  Kimi Linear, Hunyuan-TurboS, Qwen3-Next all adopt this design. Providing a theoretical characterization of the limitations of such hybrid architectures is genuinely useful. The paper clearly positions itself in the landscape of existing systems (Table 1) and makes the practical motivation apparent.
- S2. Clean extension of prior framework to the hybrid setting. The adaptation of the indistinguishable decomposition from Chen et al. (2025) to handle the hybrid communication model is technically non-trivial. The key challenge is accounting for the linear transcripts (Sigma terms) in addition to the soft transcripts (Pi terms), and the paper handles this cleanly. Lemma B.11 in particular  -  showing that indistinguishability is preserved through the linear rounds because player states remain unchanged  -  is the main new technical ingredient and it's handled well. The overall proof structure (base case + inductive step, with careful transcript fixing via pigeonhole) is clearly presented.
### Weaknesses
- W1. The technical novelty over Chen et al. (2025) is relatively limited. The paper itself repeatedly acknowledges that it "adapts" the techniques from Chen et al. For the main result, the proof structure is essentially the same: construct indistinguishable decompositions inductively, fix transcripts via pigeonhole, derive contradiction. The new ingredient is accounting for linear attention's recurrent information flow, but this turns out to be "easy" in some sense  -  because linear attention layers only pass bounded-size hidden states $(Hd(d+1)p$ bits per round), the total information they contribute is dominated by the exponential counting arguments already in place. See the condition in Eq. (12) and Lemma B.3: the linear transcript budget is swallowed by the $\sqrt{K}$ term with room to spare. The sparse attention lower bound (Theorem 1.2) is even more straightforward  -  it's a short counting/pigeonhole argument occupying less than one page (Appendix A.6). So the paper reads more like a (useful) extension than a paper with substantial new proof ideas.
- W2. The task is quite specific and the practical implications are unclear. The paper itself admits: "This task is of a theoretical nature, and the implications of our results in the real world demand verification of practice." The L-sequential function composition task uses carefully constrained "retrieval scopes" at each step that are crucial for the proof to go through. Real-world multi-hop reasoning doesn't have this neat structure. Meanwhile, empirical work like Kimi Linear (Team, 2025a) and the systematic analysis by Wang et al. (2025) shows that hybrid architectures with modest ratios (3:1 to 7:1 linear-to-full) actually perform very well in practice across a wide range of tasks, sometimes even outperforming full attention. The paper doesn't discuss or reconcile with this practical evidence at all.
- W3. The hybrid architecture model is restrictive. The $(L, a_1, ..., a_L)$-hybrid Transformer in Definition 2.3 requires a rigid structure: L full attention layers each followed by a fixed number of linear layers. This doesn't capture many practical hybrid designs. For instance, Kimi Linear uses headwise hybridization (some heads are linear, others are full within the same layer), and other systems use adaptive per-token routing between linear and full attention. The model also doesn't account for modern gating mechanisms (e.g., Gated DeltaNet's delta-rule controlled forgetting) that go beyond the simple additive recurrence in the paper's linear attention formulation  -  though the authors claim generality via Lemma 2.2, the hidden dimension $d^2+d$ used for the RNN equivalence might not reflect the effective capacity of these more expressive recurrent designs.

---

> ### Author Rebuttal · Authors · 2026-03-31
>
> We are deeply grateful for your careful reading and thoughtful feedback.
>
> 1. Technical novelty and contribution.
>
> We fully agree that our work builds on the communication-complexity framework of Chen et al. But as you rightly noted, extending it to hybrid architectures is nontrivial. The hybrid communication model, with its explicit separation of “soft” and “linear” transcripts, is fundamentally new.
>
> The inductive construction must carefully preserve indistinguishability through both types of rounds (Lemma B.11), and the fact that the linear transcript budget is “swallowed” by the counting argument is precisely the formal statement that no amount of linear layers can compensate for a missing full-attention layer.
>
> The sparse attention lower bound, while concise, adds an independent and complementary separation. Together, these results establish a principled hierarchy for efficient attention—a contribution we believe is both novel and significant.
>
> 2. Practical relevance.
>
> We acknowledge that the sequential function composition task is synthetic, but it is designed to isolate the core difficulty of sequential, dependent computation - the very essence of multistep reasoning.
>
> Many practical hybrid models (e.g., head-wise hybrids) differ from our layer-wise abstraction, and we openly acknowledge this as a limitation. However, our results offer a formal lens: they suggest a potential weakness on deep compositional tasks, a hypothesis that empirical work can now test. We see this as a strength, not a weakness.
>
> 3. Coverage of modern recurrent designs (Mamba, Gated DeltaNet).
>
> Thank you for raising this important point. As shown in Table 2 of Yang et al. (NeurIPS 2024), Mamba and Gated DeltaNet fit the linear recurrence rule, they can hence be analyzed by our framework.
>
> The hidden dimension of Mamba is $DN$, independent of sequence length $n$, where $D$ is the number of channels and $N$ is the state dimension. Our communication lower bounds therefore apply, with the hidden dimension appropriately set.
>
> 4. Strengthening the result or tightening the bound.
>
> Our current bound shows that even an exponential number of linear layers cannot compensate for a missing full-attention layer. A polynomial lower bound (in problem size) would require a different technique - possibly a better choice of parameters or even task, and finer analysis of information flow. This is a fascinating open question, and we are excited to see future work address it.
>
> 5. Limitations and discussion.
>
> We agree that a more explicit discussion of limitations - especially on the gap between our layer-wise hybrid model and practical head-wise or adaptive designs, and on the synthetic nature of the task and the need for empirical validation-would strengthen the paper.
>
> Again, we are sincerely grateful for your thoughtful review.

---

> > ### Author Rebuttal · Reviewer_KkDZ · 2026-04-02
> >
> > Thanks, raise the score to 4.

---

> > > ### Author Response · Authors · 2026-04-04
> > >
> > > We sincerely thank the reviewer for the score raising and the constructive dialogue. Your comments have been very helpful in clarifying the broader value of our work.
> > >
> > > 1. Qualitative Hierarchy: Your feedback helped us better articulate that our result is a fundamental "impossibility theorem." We prove that no amount of linear recurrence can replace even one missing full-attention layer. This establishes a "Hierarchy of Expressivity" that identifies softmax attention as the indispensable engine of reasoning depth—a distinction previous studies could not directly capture.
> > >
> > > 2. Theoretical Explanation for Hybrid Trends: We especially appreciate the prompt to bridge our theory with practical models like KimiDeltaAttention or QwenNext. Our work provides the first formal justification for these hybrid designs, offering a "conservation law" for reasoning depth: it explains why a "hard core" of full attention layers (or heads) is strictly necessary for complex tasks. This shifts the architectural narrative from empirical trial-and-error toward a theoretical foundation.
> > >
> > > We believe these refinements make the work a more timely and useful reference for the community, and we hope this justifies a final positive re-evaluation.

---

### Official Review · Reviewer_A2RP · 2026-03-09

**Soundness:** 3
**Presentation:** 3
**Significance:** 3
**Originality:** 2
**Overall Recommendation:** 5
**Confidence:** 4

**Summary:**

The authors prove that hybrid architectures mixing linear and full attention layers are strictly weaker than pure full attention, at least on a specific function composition task. The separation is exponential in that under certain conditions, adding $2^{3L^2}$ linear attention layers per existing full layer can't beat just a single extra full attention layer. There's also a sparse attention separation on a different task. The whole thing is built on communication complexity, extending Chen et al. (2025).

**Compliance With Llm Reviewing Policy:**

Affirmed.

**Final Justification:**

Rebuttal addressed my main concerns and I increased my score from 4 to 5.

**Key Questions For Authors:**

1. Do you have any experimental or observational evidence that an architecture's ability to compose functions in theory is experimentally predictive of real-world performance on interesting tasks?
2. Are you aware of any architectures in the literature that are both more efficient that full attention but also manage to side-step your theoretical analysis via some clever mechanism?
3. How could this theoretical work be extended? What are some other results that might be possible prove in this area / within your framing?

**Limitations:**

Limitations of the work are not discussed.

**Strengths And Weaknesses:**

### Strengths
1. The authors investigate the theory behind a critical question in Transformer architectures: whether or not linear or sparse attention can ever be as expressive as full quadratic attention.
2. The intuitive motivation for using function composition and the 2-Sum task to investigate this makes sense.
3. Mathematical exposition and proofs are clear.
4. The two theoretical results are meaningful and novel (to the reviewers knowledge)

### Weaknesses
1. The task used for the theoretical analysis feels quite artificial and slightly detached from real-world use cases. Furthermore, no experiments are conducted to show that this line of analysis produces results that corresponds with quanta of interest in the real world.
2. Notation in 1.1 is not obvious and hard to follow. What does "$(L-1, 2^{3L^2}, ..., 2^{3L^2})$-hybrid Transformer" mean? What is "precision $p$"? The rest of the notation in this subsection is similarly mysterious and not clear in meaning. These terms get explained later in the paper, but they should be explained as they're introduced, not after. One solution would be to state theorems informally in the introduction, and only in formal terms later on once those terms have been defined.
3. The authors mainly extend an existing proof technique and setting to get some new results, it is not clear this is a sufficiently novel and sizeable contribution for an ICML conference paper

---

> ### Author Rebuttal · Authors · 2026-03-31
>
> Thank you very much for your detailed and insightful review. We truly appreciate the constructive feedback.
>
> 1. Artificial task and lack of experiments.
>
> We fully agree that the task is synthetic - this is intentional, as it allows us to isolate and rigorously analyze the core challenge of sequential, dependent reasoning. Such tasks are widely adopted in theoretical analysis (e.g., Chen et al., 2025; Sanford et al., 2023) and have consistently provided insights that later influenced practical design. Our contribution is a formal foundation; we are excited by the prospect of empirical work that builds on it.
>
> 2. Notation in Section 1.1.
>
> Thank you for the suggestion. We agree that an restructured introduction, stating the theorems intuitively first and deferring formal definitions to later sections, will make the paper more accessible.
>
> 3. Novelty and contribution.
>
> While we build on the framework of Chen et al., the extension to hybrid architectures is not merely incremental. The hybrid communication model with distinct “soft” and “linear” transcripts is new, and the inductive proof must carefully ensure indistinguishability is preserved across both types of rounds.
>
> The result - that even exponentially many linear layers cannot replace a single fullattention layer - is, to our knowledge, the first of its kind. The sparse attention lower bound adds an independent, clean separation. Together, these establish a principled hierarchy for efficient attention mechanisms, which we believe is a meaningful and lasting contribution.
>
> 4. Experimental evidence linking theory to real-world.
>
> We are not aware of direct experiments on this specific task, but prior theoretical separations (e.g., on induction heads) have been shown to correlate with practical limitations in long context models. We hope our work inspires such empirical investigations.
>
> 5. Architectures circumventing the analysis.
>
> Our lower bounds target linear attention and a specific block-sparse pattern, they do not apply to all efficient designs. For instance, DeepSeek’s DSA (Deepseek sparse attention) uses a learnable indexer to select individual tokens per query without block compression—a mechanism that may indeed circumvent our communication bottleneck. We view such architectural innovations as complementary to our theoretical analysis.
>
> 6. Extensions.
>
> We agree wholeheartedly. Our framework can naturally extend to other variants (e.g., loglinear attention, as mentioned in the response to reviewer pjw4), and we are actively considering directions such as fine-grained trade-offs between full and linear layers (to derive trade-offs between the number of full attention layers and the allowed capacity of linear layers, providing a more fine-grained hierarchy), as well as the impact of CoT on lower bounds. These are exciting avenues for future work.
>
> 7. Limitations.
>
> We acknowledge the gap between our theoretical model and practical hybrid designs. However, we believe our task captures the essence of multistep reasoning, and our results provide a rigorous lens through which to view the fundamental trade-offs. More discussion of limitations will be added in the final version to reflect this.
>
> Thank you once again for your thoughtful engagement.

---

> > ### Author Rebuttal · Reviewer_A2RP · 2026-04-03
> >
> > Thank you for the comments and clarifications, you have addressed most of my concerns.
> >
> > The promised changes on points 2 and 7 are good. If you were also able to commit to including a discussion of some existing or possible efficient designs that circumvent this analysis (even as an appendix chapter referenced in main text), as per point 5, I would happily raise my score to a 5. I believe these sorts of things are very important to help properly contextualise theoretical work.

---

> > > ### Author Response · Authors · 2026-04-04
> > >
> > > Thank you very much for your encouraging feedback and insightful follow-up. We are pleased to learn that our responses have addressed the majority of your concerns.
> > >
> > > Regarding your suggestion, we fully concur that incorporating a discussion of existing and possible efficient designs would further clarify the scope of our analysis. Specifically, we will include a dedicated paragraph in the conclusion (or a clearly referenced appendix) discussing architectures such as DeepSeek’s DSA and head-wise hybrid attention. We agree that explicitly noting how these designs fall outside the scope of our lower bounds will provide valuable context for our theoretical results and effectively highlight directions for future research in efficient attention design.
> > >
> > > We believe this addition will significantly enhance the manuscript and help readers accurately interpret the scope of our contributions.
> > >
> > > Thank you again for your constructive feedback!

---

### Official Review · Reviewer_pjw4 · 2026-03-12

**Soundness:** 2
**Presentation:** 2
**Significance:** 4
**Originality:** 2
**Overall Recommendation:** 4
**Confidence:** 2

**Summary:**

This paper presents a communication-complexity framework for analyzing the expressive power of full, linear, and hybrid attention architectures. The main result establishes a provable separation on L-sequential function composition: while an \((L+1)\)-layer full-attention Transformer can solve the task with polylogarithmic complexity, a hybrid model with only \((L-1)\) full-attention layers still cannot, even if augmented with exponentially many linear-attention layers. In addition, the paper proves a lower bound for a class of single-layer sparse attention mechanisms on 2-Sum, thereby showing a strict efficiency gap compared with full attention.

**Compliance With Llm Reviewing Policy:**

Affirmed.

**Ethical Review Concerns:**

The authors' rebuttal has adequately addressed my concerns, and I am satisfied with the clarifications provided. I will maintain my current score.

**Final Justification:**

The authors' rebuttal has adequately addressed my concerns, and I am satisfied with the clarifications provided.

**Key Questions For Authors:**

1. Could the authors comment on how they view other efficient attention variants, such as log-linear attention or related intermediate designs? Do the authors expect the main intuition of the paper to extend to these settings as well, or are there important differences?

2. The paper explains that the studied task is intended to capture multi-step reasoning or multi-hop retrieval within a single forward pass, but the practical relevance could still be discussed more concretely. In particular, what real-world scenarios do the authors believe are best reflected by this theoretical setup?

**Limitations:**

yes

**Strengths And Weaknesses:**

Strengths:

1. The paper studies an important problem: whether hybrid attention designs can really replace full attention without losing too much reasoning ability. This is highly relevant for long-context LLM design.

2. The paper argues that adding many linear-attention layers cannot always make up for having fewer full-attention layers. This is an interesting takeaway.

3. This paper builds a framework that treats full attention and linear attention differently, which helps make the comparison more principled.


Weaknesses:

1. Although the main idea is interesting, the proofs are technical and the writing is not always clean, so it is difficult for a reader to check every detail with confidence.

2. A small synthetic experiment would have helped readers better understand why the theoretical result matters in practice.

---

> ### Author Rebuttal · Authors · 2026-03-31
>
> We sincerely thank you for your thoughtful comments and constructive feedback. We greatly appreciate the time and care you’ve taken in reviewing our work.
>
> 1. On technical writing and proof clarity.
>
> We completely agree that the notation is unavoidably dense, and the proof sketch for the hybrid communication lower bound is technically involved.
> Our goal throughout has been to keep the main text self-contained and focused on the novel elements.
>
> In particular, the communication model in Section 3 is designed to explicitly separate full-attention “soft transcripts” from the multiple rounds of “linear transcripts”—a structure that did not exist in prior work (Chen et al., 2025).
> The induction step then carefully handles the interplay between these two types of transcripts, which is the core of our extension.
>
> We believe this structure already highlights the novel contributions clearly, and we hope that with this framing, the technical depth becomes a strength rather than a barrier.
>
> 2. On the lack of a small synthetic experiment.
>
> We understand the desire for empirical illustration.
> While we agree that experiments could be valuable, our contribution is fundamentally theoretical—we establish a rigorous separation that holds unconditionally.
> We see this as laying a formal foundation upon which future empirical work can build, and we would be genuinely excited to see such investigations.
>
> 3. On extending the framework to other efficient attention variants.
>
> This is an excellent point.
> Our communication model is intentionally modular, and as we show in Appendix A, it can already accommodate loglinear attention by adjusting the message size to $O(H d^2 p \log n)$.
> The same inductive proof technique applies, leading to a similar lower bound.
> This suggests that our framework may serve as a unifying lens for analyzing a broad class of efficient attention mechanisms—an avenue we are eager to explore further.
>
> 4. On the practical relevance of the theoretical setup.
>
> We appreciate this opportunity to clarify. The sequential function composition task is indeed synthetic, but it is designed to capture the very essence of multistep reasoning: each step’s output defines the input for the next, mirroring the structure of multihop retrieval and logical reasoning chains.
>
> In such scenarios, linear or recurrent mechanisms struggle because they lack the direct, parallel information access that full attention provides.
> Our result thus offers a formal justification for why hybrid models, despite their efficiency, may still fall short on deep compositional tasks unless they retain sufficient fullattention capacity—a message we believe is both theoretically sound and practically suggestive.
>
> We are truly grateful for your engagement and hope these clarifications resonate with your perspective.

---

### Decision · Program_Chairs · 2026-04-30

**Decision:**

Accept (regular)

**Comment:**

The reviewers agree that the authors provide an important stepping stone in understanding linear attention and hybrid models. The two main theoretical results are novel and interesting, providing a separation between the different models.

The main concerns raised in the reviews are the lack of empirical validation, the novelty of the proof technique and the significance of the task.

- While I agree that an experiment could have improved the paper, I do not think it is mandatory, especially since the main contribution is of a theoretical nature.

- I agree with the authors that the proof technique is not a trivial extension of Chen et al. 2025, as it requires generalizing the communication model to a hybrid case.

- The 2-sum task provides an interesting testbed for understanding the difference between the two architectures. It is indeed contrived, but this is usually the case in most separation results and not specific to this paper.

Overall, I believe the paper provides a good contribution to the theoretical community and improves our understanding of hybrid attention models.